# Above-room-temperature chiral skyrmion lattice and Dzyaloshinskii–Moriya interaction in a van der Waals ferromagnet Fe$_{3-x}$GaTe$_2$

Chenhui Zhang [1,5], Ze Jiang[2,5], Jiawei Jiang[3,5], Wa He[2], Junwei Zhang [2], Fanrui Hu[1], Shishun Zhao[1], Dongsheng Yang[1], Yakun Liu[1], Yong Peng[2] ✉, Hongxin Yang [3,4] ✉ & Hyunsoo Yang [1] ✉

Skyrmions in existing 2D van der Waals (vdW) materials have primarily been limited to cryogenic temperatures, and the underlying physical mechanism of the Dzyaloshinskii–Moriya interaction (DMI), a crucial ingredient for stabilizing chiral skyrmions, remains inadequately explored. Here, we report the observation of Néel-type skyrmions in a vdW ferromagnet Fe$_{3-x}$GaTe$_2$ above room temperature. Contrary to previous assumptions of centrosymmetry in Fe$_{3-x}$GaTe$_2$, the atomic-resolution scanning transmission electron microscopy reveals that the off-centered Fe$_{II}$ atoms break the spatial inversion symmetry, rendering it a polar metal. First-principles calculations further elucidate that the DMI primarily stems from the Te sublayers through the Fert–Lévy mechanism. Remarkably, the chiral skyrmion lattice in Fe$_{3-x}$GaTe$_2$ can persist up to 330 K at zero magnetic field, demonstrating superior thermal stability compared to other known skyrmion vdW magnets. This work provides valuable insights into skyrmionics and presents promising prospects for 2D material-based skyrmion devices operating beyond room temperature.

Topological magnetic excitations such as magnetic skyrmions have gained enduring research interest in the field of spintronics for more than one decade[1–3]. Typically, magnetic skyrmions are observed in chiral magnets and heavy metal/ferromagnet multilayers[4,5], where either bulk or interfacial inversion symmetry breaking induces the Dzyaloshinskii–Moriya interaction (DMI), favoring the helical magnetic order. The interplay of DMI, exchange interaction, magnetic anisotropy, and external magnetic fields results in the delicate spin textures of skyrmions. These particle-like topological entities are expected to act as information carriers that can be written, erased, and driven by external stimuli with low energy consumption[6]. In addition, certain degrees of freedom in skyrmions, including helicity and polarity, are switchable[7,8], making them particularly suitable for logic and memory applications. In the meantime, the discovery of 2D van der Waals (vdW) magnetic materials, such as Cr$_2$Ge$_2$Te$_6$[9], CrI$_3$[10], and Fe$_3$GeTe$_2$[11], has stimulated a new upsurge in spintronics and materials research. These 2D vdW magnets provide an extraordinary platform for exploring fundamental physics, novel device architectures, and exotic quantum and topological phases[12–14]. Notably, skyrmions have been successfully demonstrated in some vdW

[1]Department of Electrical and Computer Engineering, National University of Singapore, Singapore 117576, Singapore. [2]School of Materials and Energy and Electron Microscopy Centre of Lanzhou University, Lanzhou University, Lanzhou 730000, China. [3]National Laboratory of Solid State Microstructures, School of Physics, Collaborative Innovation Center of Advanced Microstructures, Nanjing University, Nanjing 210093, China. [4]Center for Quantum Matter, School of Physics, Zhejiang University, Hangzhou 310058, China. [5]These authors contributed equally: Chenhui Zhang, Ze Jiang, Jiawei Jiang. ✉e-mail: pengy@lzu.edu.cn; hongxin.yang@zju.edu.cn; eleyang@nus.edu.sg

ferromagnets[15–18], opening up new avenues for 2D topological magnetism.

However, the skyrmion phase has generally been limited to cryogenic temperatures in vdW ferromagnets, which significantly restricts their practical applications. For instance, magnetic skyrmions were typically observed below 60 K in $Cr_2Ge_2Te_6$[18,19] and 185 K in $Fe_3GeTe_2$[20–26]. Recently, (nearly) room-temperature skyrmions were reported in $Fe_5GeTe_2$[16,27] and $Cr_xTe_2$[28,29]. Note that these skyrmions should be classified as dipole skyrmions (or skyrmion bubbles/type-I bubbles), which are primarily stabilized by the dipolar and exchange interactions[30]. Compared to the dipole skyrmions, the DMI-involved chiral skyrmions commonly have a smaller size[30], making them more favorable for high-density device integration. In addition, the DMI is the key ingredient that breaks the energy degeneracy between opposite chiralities, leading to the formation of homochiral skyrmions and other topological spin textures[4,31]. Homochirality ensures a consistent motion direction when skyrmions are driven by a spin-polarized current. More interestingly, by modulating the DMI to create an ideal Bloch-Néel hybridization, a zero skyrmion Hall angle may be achieved[32]. These DMI-mediated features can be very useful for skyrmion-based racetrack memories. However, unfortunately, most of the vdW magnets possess centrosymmetric crystal structures[33]. Although the inversion symmetry breaking and chiral skyrmions can possibly be induced by, for example, self-intercalation[34,35], doping[17], and constructing Janus 2D magnets[36], these approaches are not easy to achieve. For instance, though Néel skyrmions are observed in self-intercalated chromium telluride[34], more studies report Bloch-type achiral skyrmions and biskyrmions even with similar intercalate concentrations[28,29,37], implying that the asymmetric intercalation is challenging. $(Fe_{0.5}Co_{0.5})_5GeTe_2$ is a remarkable material that can host Néel skyrmion lattice up to 312 K at zero magnetic field[17,38]. To obtain the high-temperature ferromagnetic order, the key factors are precisely regulating the Co-doping concentration at ~50% and acquiring the $AA'$-type layer stacking[39,40]. Nevertheless, the stacking mechanism is so far unclear, and recent studies also observe the antiferromagnetic order in $(Fe_{0.5}Co_{0.5})_5GeTe_2$ due to the existence of $AA$-type stacking mode[41]. On the other hand, the DMI mechanism in these 2D magnetic materials has yet to be deeply understood.

In this work, we propose a new vdW chiral-skyrmion-hosting material $Fe_{3-x}GaTe_2$ ($x \approx 0.15$). Contrary to the prior knowledge of centrosymmetric $Fe_{3-x}GaTe_2$, we demonstrate that the off-centered $Fe_{II}$ atoms can break the inversion symmetry and induce a robust DMI under the scheme of Fert−Lévy model[42,43]. We also find that the Néel skyrmion lattice phase can survive up to 330 K, which is higher than all other known skyrmion vdW magnets.

## Results

### Characterizations of $Fe_{3-x}GaTe_2$

The $Fe_{3-x}GaTe_2$ single crystals used in this study are synthesized by using a modified chemical vapor transport (CVT) method (Supplementary Note 1). The magnetic properties of bulk crystals are investigated using a vibrating sample magnetometer (VSM). The $(00l)$ peaks in the $\theta-2\theta$ X-ray diffraction spectrum (Supplementary Fig. 2) imply that the crystallographic $c$ axis is along the out-of-plane direction of the crystal. Figure 1a shows the temperature-dependent magnetization of $Fe_{3-x}GaTe_2$ under the in-plane ($H\perp c$, blue curves) and out-of-plane ($H//c$, red curves) magnetic field configurations, where $H\perp c$ and $H//c$ indicates the field is perpendicular and parallel to the $c$ axis, respectively. Both zero-field-cooled (ZFC) and field-cooled (FC) protocols are adopted, which correspond to the dashed and solid lines, respectively. One can observe that the out-of-plane magnetization is significantly larger than the in-plane one, indicating a perpendicular magnetic anisotropy (PMA). By plotting the derivative of the out-of-plane FC curve, we can extract the Curie temperature ($T_c$) of the $Fe_{3-x}GaTe_2$ bulk crystal as 350 K (inset of Fig. 1a), which is much higher than the other Fe-based vdW magnets $Fe_\delta GeTe_2$ ($\delta = 3, 4, 5$)[15,44,45]. The bulk isothermal magnetization results are presented in Fig. 1b and Supplementary Fig. 3, corresponding to the $H//c$ and $H\perp c$ configurations, respectively. The PMA is further verified by the fact that $Fe_{3-x}GaTe_2$ is much easier to be saturated under the out-of-plane field. It is interesting to note that the out-of-plane magnetization curves below $T_c$ show two distinct slopes (i.e. the steep and slanted slopes), which is caused by the irreversible process corresponding to the nucleation and annihilation of domains[46]. In the saturation process, the stripe domains fragment into short chains and skyrmions, leading to a gradual increase of magnetization until saturation. Whereas in the demagnetization process, the

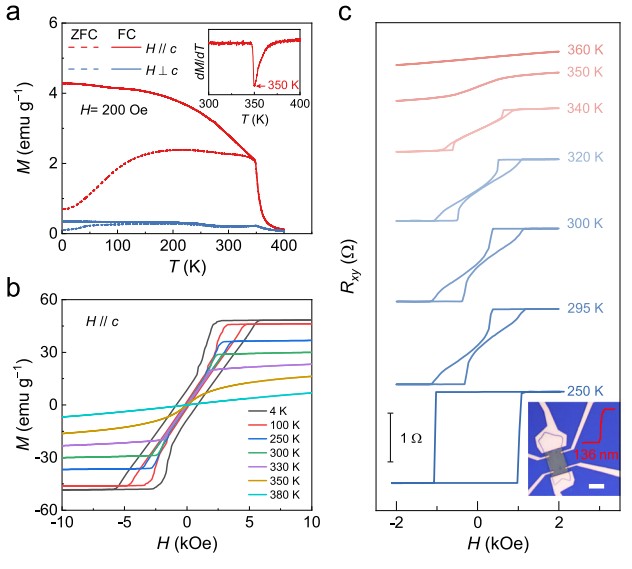

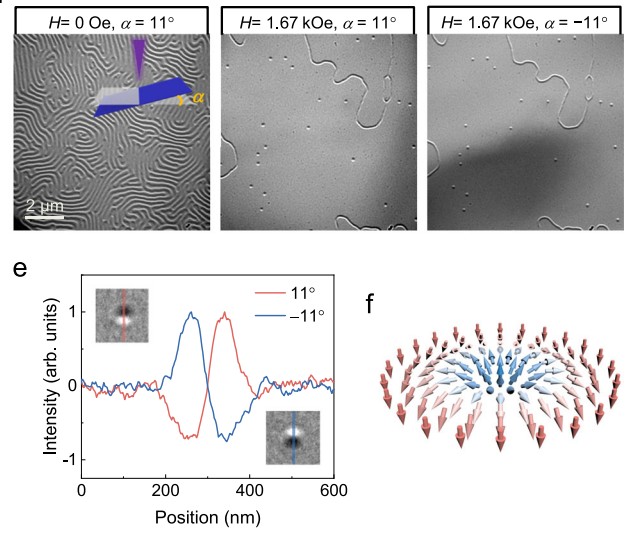

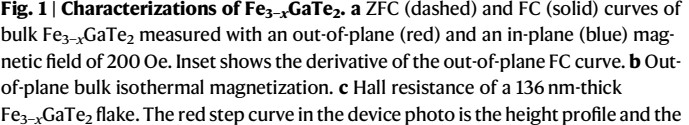

**Fig. 1 | Characterizations of $Fe_{3-x}GaTe_2$. a** ZFC (dashed) and FC (solid) curves of bulk $Fe_{3-x}GaTe_2$ measured with an out-of-plane (red) and an in-plane (blue) magnetic field of 200 Oe. Inset shows the derivative of the out-of-plane FC curve. **b** Out-of-plane bulk isothermal magnetization. **c** Hall resistance of a 136 nm-thick $Fe_{3-x}GaTe_2$ flake. The red step curve in the device photo is the height profile and the scale bar is 10 μm. **d** L-TEM images of labyrinth domains (left) and field-induced skyrmions (middle and right) in a 179 nm-thick $Fe_{3-x}GaTe_2$ flake. The sample is ZFC from 370 to 295 K before imaging. **e** Intensity profile of a skyrmion under reversed tilt angles. **f** Schematic of a Néel-type skyrmion.

single-domain state remains until the labyrinth domain pops up, leaving a steep slope on the magnetization curve. Similar sheared loops are also observed in the Hall resistance of exfoliated $Fe_{3-x}GaTe_2$ flakes, as shown in Fig. 1c. In many skyrmion-hosting materials, including magnetic multilayers[47,48] and single crystals[17,22], labyrinth domains usually emerge in the slanted area and they can further transform into skyrmions by tuning the magnetic field.

Since the characteristic sheared hysteresis loops give hints on the emergence of magnetic skyrmions in $Fe_{3-x}GaTe_2$, we next use the Lorentz transmission electron microscopy (L-TEM) to examine the domain structures. The L-TEM samples are mechanically exfoliated from bulk crystals and subsequently transferred onto a silicon nitride TEM grid. Before imaging, the 179 nm-thick flake sample (see the atomic force microscopy image in Supplementary Fig. 4) is ZFC from 370 to 295 K. At zero magnetic field, labyrinth domains are observed, as shown in Fig. 1d. Intriguingly, some spot domains emerge with the gradual increase of the out-of-plane magnetic field. These spot domains have a half-dark and half-bright contrast. More importantly, the contrast is reversed when the sample is tilted with positive and negative angles (Fig. 1e), and it becomes invisible without tilting (Supplementary Fig. 5). These behaviors are consistent with the features of Néel-type skyrmions (Fig. 1f) under the L-TEM[47]. With increasing the magnetic field, more skyrmions emerge. Whereas they start to decrease when the magnetic field is larger than 1.59 kOe and finally disappear after saturation (Supplementary Fig. 6).

More interestingly, we also observe the magnetic skyrmionium state in $Fe_{3-x}GaTe_2$, as pointed out by the green arrows in Supplementary Fig. 6. The skyrmionium consists of a uniformly magnetized core region encircled by a 360° domain wall loop[49]. Unlike skyrmions that have a unitary topological charge, skyrmioniums show zero topological charge[49]. As skyrmioniums do not exhibit the skyrmion Hall effect when driven by a spin polarized current, they are considered promising candidates for racetrack memory applications[50]. Moreover, the skyrmionium is observed to transform into a skyrmion when the magnetic field increases from 1.47 to 1.59 kOe (Supplementary Fig. 6f). Similar phenomena are also observed in $Fe_{3-x}GeTe_2$[51] and $Cr_2Ge_2Te_6$[52] at low temperatures. The diversity of topological spin textures and their phase transitions reveal that $Fe_{3-x}GaTe_2$ is a rich platform for room-temperature skyrmionics.

## Above-room-temperature Néel skyrmion lattice at zero magnetic field

We have demonstrated that $Fe_{3-x}GaTe_2$ is a new room-temperature skyrmion-hosting vdW magnet. However, as displayed in Fig. 1d, the skyrmions obtained merely by tuning the magnetic field from the demagnetized state always coexist with stripe domains and the skyrmion density is quite low. Compared to isolated skyrmions, the skyrmion lattice is more attractive for high-density storage applications. To obtain a dense skyrmion lattice, a specific FC procedure is adopted. As schematically shown in Supplementary Fig. 7a, the sample is first applied with an out-of-plane magnetic field of 360 Oe at 370 K (above $T_c$); afterwards, it is cooled down to 295 K with the magnetic field. Generally, in many bulk helimagnets and vdW ferromagnetic thin flakes such as $Fe_3GeTe_2$ and $(Fe_{0.5}Co_{0.5})_5GeTe_2$, the equilibrium skyrmion lattice phase is confined to a small $H$-$T$ region just below $T_c$[4,17,26]. However, once the sample is cooled from the paramagnetic state with a suitable magnetic field and passes through this phase pocket, the skyrmion lattice emerges and it can persist even outside of this region due to the topological stability. As expected, closely-packed skyrmions are successfully created at 295 K, as shown in Supplementary Fig. 7b.

More interestingly, the skyrmion lattice still survives after the external field is removed. Figure 2a–c exhibit the L-TEM images of skyrmions captured with positive, zero, and negative tilt angles at zero

magnetic field, respectively. One can observe two different types of particle domains, which are the dark–bright small spots (red circles in Fig. 2d, e) and the dark–bright–dark–bright large spots (blue circles in Fig. 2f, g). At first glance, the dark–bright–dark–bright spots look like the type-II Bloch bubbles[16]. Nevertheless, these particles do not show any contrast without sample tilting (Fig. 2b), indicating that they are Néel-type spin textures. Figure 2h displays the simulated L-TEM images of a small-sized (up) and a large-sized (down) Néel-type skyrmion with different tilt angles. It is evident that the contrast of the Néel-type skyrmion can change from a dark–bright to a dark–bright–dark–bright spot when it becomes larger in size. Hence, the small and large particle domains obtained here are both Néel-type skyrmions. Similar Néel skyrmion images with additional contrast inside are also observed in some other ferromagnets such as $Cr_{1.3}Te_2$[34] and $PtMnGa$[53]. In Supplementary Fig. 8, we perform more L-TEM experiments on the defocus- and tilt-angle-dependent imaging of Néel-type skyrmions in $Fe_{3-x}GaTe_2$. One can observe that the magnetic contrast is gradually enhanced with the increase of defocus at a fixed tilt angle. The additional contrast inside the skyrmion is greatly suppressed when the defocus decreases to ± 0.5 mm, accompanied by a significant compromise of magnetic contrast at the same time. Therefore, the additional contrast in some Néel skyrmions should be caused by the relatively large skyrmion size and the large defocus that is used to enhance the magnetic contrast.

Figure 3a shows the thermal stability of the metastable skyrmion lattice at zero magnetic field in $Fe_{3-x}GaTe_2$. When the temperature increases from 295 to 330 K, the skyrmion lattice is robust enough to resist elevated thermal perturbation. At the same time, the skyrmion size slightly decreases, resulting in a gradual increase of skyrmion density up to ~21 $\mu m^{-2}$ at 330 K. The decrease of skyrmion size could be due to the decreasing saturation magnetization with increasing temperature[54]. Nevertheless, the skyrmion lattice transforms into stripe domains when the temperature reaches 340 K. This can be understood that the metastable skyrmion state stays at the local energy minima after the FC process, whereas the thermal perturbation at 340 K is so strong that it overcomes the energy barrier and pushes the system into the stripe domain state (global energy minimum). Hitherto, closely-packed skyrmion lattices have been created in some vdW and quasi-vdW magnetic materials at zero magnetic field utilizing different strategies, such as the FC process[15,23,29,34], the field training by magnetic tips[17], and the current-induced thermal effect[22], as summarized in Fig. 3b. One can see that $Fe_3GeTe_2$ can form high-density metastable skyrmions at zero magnetic field, whereas they only survive at low temperatures. In this work, we demonstrate that $Fe_{3-x}GaTe_2$ possesses a record-high critical temperature (between 330 to 340 K) of skyrmion lattice state at zero magnetic field among known vdW magnets. Even in traditional non-vdW bulk materials, high-temperature chiral skyrmions are scarce. For instance, $GaV_4S_8$, $VOSe_2O_5$, and $PtMnGa$ are polar magnets hosting Néel-type skyrmions, whereas their skyrmion phase emerges far below room temperature[53,55,56]. β-Mn-type Co-Zn-Mn alloy and B20-type $Co_{1.043}Si_{0.957}$ are two rare examples that can host chiral skyrmions at room temperature but in Bloch-type[57–59]. While in some antiskyrmion compounds, such as $Mn_{1.4}Pt_{0.9}Pd_{0.1}Sn$ and $Fe_{1.9}Ni_{0.9}Pd_{0.2}P$, the antiskyrmion phase can reach as high as 400 K[60,61]. It is worth mentioning that, in current-driven skyrmion motion, Joule heating is inevitable because a large current density is required to overcome the pinning and achieve high-speed motion, which can cause a temperature rise of several tens of kelvin[62]. Therefore, skyrmion materials with a skyrmion phase temperature just reaching room temperature may not be able to function as practical devices at room temperature. The observation of above-room-temperature high-density chiral skyrmion lattice at zero magnetic field in $Fe_{3-x}GaTe_2$ opens promising perspectives for developing novel spintronic devices based on 2D magnets.

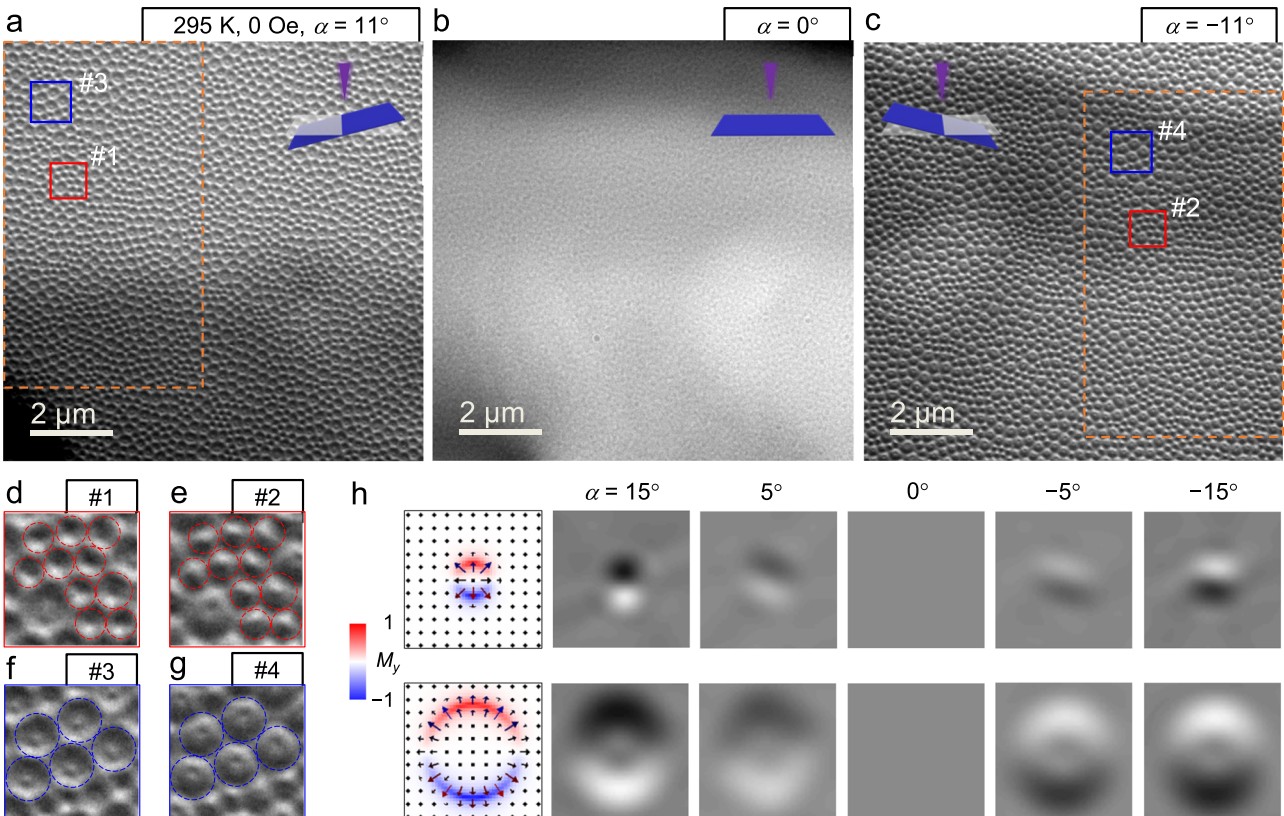

**Fig. 2 | Room-temperature metastable skyrmion lattice at zero magnetic field.**
**a**–**c** L-TEM images of a 179 nm-thick Fe$_{3-x}$GaTe$_2$ flake under different tilt angles. Before imaging, the sample is FC from 370 to 295 K under an out-of-plane magnetic field of 360 Oe. For clarity, the identical area in **a** and **c** is outlined by orange dashed lines. The defocus value is −3 mm. **d**–**g** Magnified images of four regions extracted from **a** and **c**. The red and blue circles outline two different types of spot domains. **h** Simulated L-TEM images of a small-sized (up) and a large-sized (down) Néel-type skyrmion under different tilt angles.

## DMI in Fe$_{3-x}$GaTe$_2$

For complicated or arbitrary magnetic domain patterns, the average surface domain width $w$ can be estimated using a stereological method[63]: $w = \frac{2}{\pi} \frac{\sum_i l_i}{\sum_i n_i}$, where $l_i$ is the length of the $i$th test line and $n_i$ is the number of intersections of the $i$th test line with the domain walls. In Supplementary Fig. 9, five test lines are randomly chosen, which give an average domain width $w \sim 0.24 \pm 0.01$ μm at room temperature. Then, according to the phenomenological model[63], the average domain wall energy $\delta_w$ can be calculated as $\delta_w = \frac{wM_s^2}{4\pi\beta}$, where $M_s$ is the saturation magnetization and $\beta$ is a phenomenological fitting parameter that is approximately equal to 0.31 for magnets with high magnetocrystalline anisotropy. Therefore, the average domain wall energy at room temperature is estimated to be 0.22 mJ m$^{-2}$.

Alternatively, the domain wall energy can be extracted via micromagnetic analysis. When the DMI is included, the domain wall energy is expressed as[64] $\delta_w = 4\sqrt{AK_{eff}} - \pi|D|$, where $A$ is the exchange stiffness constant and $K_{eff}$ is the effective magnetic anisotropy constant. They are estimated to be 0.70 pJ m$^{-1}$ and 0.30 MJ m$^{-3}$ at room temperature, respectively (Supplementary Note 2). Without considering the DMI, the first term $4\sqrt{AK_{eff}}$, which is the domain wall energy for non-DMI systems, is calculated to be 1.83 mJ m$^{-2}$. This value is greatly larger than $\delta_w$ obtained using the phenomenological model (i.e. 0.22 mJ m$^{-2}$), indicating that the DMI term plays a significant role in Fe$_{3-x}$GaTe$_2$. Thus, the DMI constant is estimated to be $|D| = 0.51$ mJ m$^{-2}$. More detailed analysis can be found in Supplementary Note 2.

The emergence of DMI and Néel-type skyrmions is inconsistent with the prior knowledge of the centrosymmetric structure of Fe$_{3-x}$GaTe$_2$[65]. Considering that the exfoliated flakes gradually oxidize in ambient conditions, there would be an oxide layer on the surface. In another vdW magnet Fe$_3$GeTe$_2$, first-principles calculations revealed that the interface between the pristine and oxidized Fe$_3$GeTe$_2$ could break the spatial inversion symmetry and generate a considerable DMI, which might be responsible for the Néel-type skyrmions[22]. To determine whether the DMI in Fe$_{3-x}$GaTe$_2$ is from the oxide interface, we perform a control experiment in which the L-TEM samples are prepared in a N$_2$-filled glovebox and covered with hBN before exposed to ambient air. Supplementary Fig. 11 shows a 130 nm-thick Fe$_{3-x}$GaTe$_2$ flake which is partially protected by hBN. Electron energy loss spectroscopy (EELS) measurements reveal that the hBN layer can effectively protect Fe$_{3-x}$GaTe$_2$ from oxidization (Supplementary Fig. 12). In the ZFC experiments, labyrinth domains are observed in both the exposed and hBN-protected regions, as shown in Supplementary Fig. 13. The domain walls are determined to be Néel-type as they do not show contrast without tilting. When the magnetic field increases to 1.02 kOe, several isolated skyrmions emerge in both regions (Supplementary Fig. 14). Additionally, in the FC experiments, we again successfully create dense skyrmions at 300 K at zero magnetic field, as shown in Supplementary Fig. 15, and they are also demonstrated to be Néel-type. With increasing the magnetic field, the skyrmion density gradually decreases (Supplementary Fig. 16). It is noteworthy that there is no noticeable difference in the skyrmion type, size, and density between the exposed and hBN-protected Fe$_{3-x}$GaTe$_2$, indicating that the oxide surface does not considerably enhance the DMI in the present case. Thus, the DMI should be intrinsically from Fe$_{3-x}$GaTe$_2$ itself.

## Discussion

In order to find out the origin of DMI, we investigate the crystal structure of Fe$_{3-x}$GaTe$_2$ using aberration-corrected high-angle annular

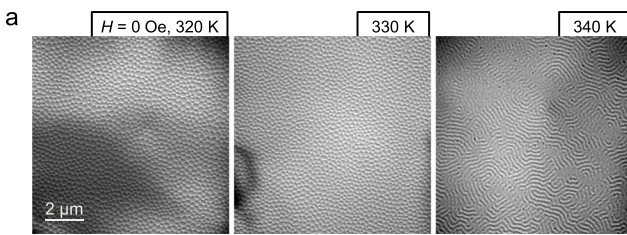

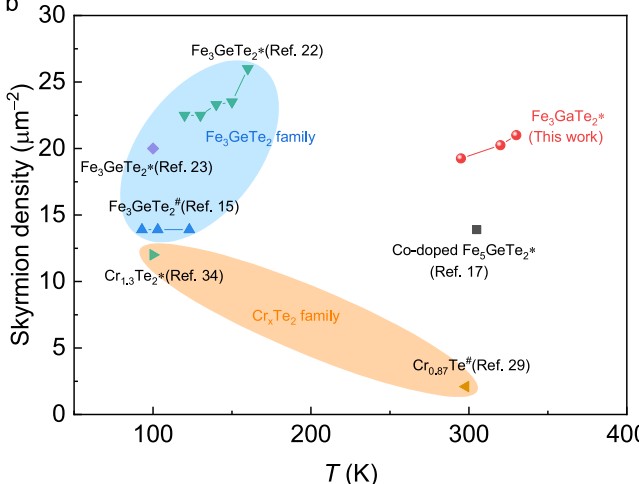

**Fig. 3 | Above-room-temperature skyrmion lattice at zero magnetic field in Fe$_{3-x}$GaTe$_2$. a** Temperature dependence of skyrmion lattice in a 179 nm-thick Fe$_{3-x}$GaTe$_2$ flake. The sample is initially FC from 370 to 295 K under an out-of-plane magnetic field of 360 Oe. The defocus value is −3 mm. **b** Comparison of skyrmion lattices at zero magnetic field in various vdW and quasi-vdW magnets[15,17,22,23,29,34]. The markers # and * indicate Bloch-type skyrmion bubbles and Néel-type skyrmions, respectively.

dark-field scanning transmission electron microscopy (HAADF-STEM). Figure 4a, d present the atomic arrangement along the [100] and [001] zone axis, respectively. In these Z-contrast micrographs, the intensity is approximately proportional to the square of atomic number (~$Z^2$), which allows one to distinguish different atom columns by intensity. Compared to the formerly reported $P6_3/mmc$ structure[65], unexpected vertical displacements are observed at Fe$_{II}$ sites (orange balls) in both upper and lower layers of the unit cell and they are along the same direction, as shown in Fig. 4b, c and Supplementary Fig. 17. Whereas in the *ab* plane (Fig. 4d), no atomic displacement is detected, indicating that the Fe$_{II}$ atoms only shift along the *c* axis. The vertical displacements of Fe$_{II}$, assuming their values are identical in the two layers, can break the centrosymmetry and lead to $P6_3mc$ space group. Nevertheless, a small discrepancy of the displacements between the two layers can further lower the symmetry to the subgroup $P3m1$. Our HAADF-STEM results support such discrepancy (see Fig. 4c and Supplementary Fig. 17).

To further determine the crystal structure on a larger scale, we perform single-crystal X-ray diffraction (SCXRD) using a millimeter-sized crystal. Based on the aforementioned discussion, we use these three space groups ($P6_3/mmc$, $P6_3mc$, and $P3m1$) to perform the structure refinement, which lead to goodness of fit of 1.376, 1.305, and 1.116, respectively. Thus, $P3m1$ provides the best fit quality. The refined crystallography data are summarized in Supplementary Tables 1 & 2. The simulated HAADF-STEM images based on the refined structure are consistent with the experimental observations, as shown in Supplementary Fig. 18. Moreover, we perform selected-area electron diffraction (SAED) on Fe$_{3-x}$GaTe$_2$ lamellar samples along [100] and [210] zone axes. It is known that for $P6_3/mmc$ and $P6_3mc$ space groups, the reflection condition should satisfy (*hhl*), where *l* is an even number (i.e.

*l* = 2*n* and *n* is an integer)[66], however $P3m1$ has no such extinction rule. In Supplementary Fig. 19, the (001), (003), and (005) reflection spots are unambiguously identified, indicating that Fe$_{3-x}$GaTe$_2$ should belong to $P3m1$ rather than $P6_3/mmc$ or $P6_3mc$. Thus, the SAED patterns are consistent with the HAADF-STEM and SCXRD results. The schematic of Fe$_{3-x}$GaTe$_2$ crystal structure is illustrated in Fig. 4e. The off-centered Fe$_{II}$ atoms lower the crystal symmetry, making Fe$_{3-x}$GaTe$_2$ a polar metal that belongs to the non-centrosymmetric space group $P3m1$ (point group $C_{3v}$).

The determination of Fe$_{3-x}$GaTe$_2$ structure allows us to further explore the physical origin of DMI at the atomic scale. In each Fe$_{3-x}$GaTe$_2$ monolayer, the middle Fe$_I$, Fe$_{II}$, and Ga atoms are sandwiched by two heavy metalloid Te sublayers. The interface between the Te and Fe$_I$ sublayers may provide an interfacial DMI through the Fe$_I$–Te–Fe$_I$ path. Following the Moriya symmetry rules[67], the DMI vector is in the form of $\mathbf{d}_{ij} = d(\hat{\mathbf{z}} \times \hat{\mathbf{u}}_{ij})$, where *d* is the microscopic DMI constant, and $\hat{\mathbf{z}}$ and $\hat{\mathbf{u}}_{ij}$ are unit vectors pointing normal to the interface and from site *i* to *j*, respectively. Because the reflection symmetry of the Fe$_I$–Te sublattice in each Fe$_{3-x}$GaTe$_2$ monolayer, the DMI contributed by the up and down Fe$_I$ sublayers cancel each other out. This conclusion is also valid for the Fe$_{II}$ sublayers when they are at the middle position. However, once the Fe$_{II}$ displacement breaks the symmetry, a net DMI is expected to arise. The second-order DMI tensor for crystallographic point groups can be derived by applying Neumann symmetry principle of crystallography and generating matrices[68], which can be expressed as $\mathbf{D}_{ij} = |\sigma| \sigma_{ii'} \sigma_{jj'} \mathbf{D}_{i'j'}$, where $\sigma$ contains all the symmetry operations contained in the specific point group. The nonzero diagonal or off-diagonal entries in the matrix determine the type of DMI, i.e. bulk or interfacial. For Fe$_{3-x}$GaTe$_2$ with Fe$_{II}$ displacement, the space group reduces to $P3m1$ ($C_{3v}$). We can identify the DMI tensor of $C_{3v}$ point group:

$$\mathbf{D}(C_{3v}) = \begin{pmatrix} 0 & D_{12} & 0 \\ -D_{12} & 0 & 0 \\ 0 & 0 & 0 \end{pmatrix} \quad (1)$$

with one independent nonzero off-diagonal entries $D_{12}$. This essentially ensures the existence of interfacial-type DMI, while bulk-type DMI is absent under this condition.

Based on the above analysis, density functional theory (DFT) based first-principles calculations are adopted to quantitatively evaluate the DMI in Fe$_{3-x}$GaTe$_2$ (details in Supplementary Note 3). Using the real-space spin spiral method[69,70], the DMI parameter *d* can be calculated from the energy difference between the clockwise (CW) and counter-clockwise (CCW) Néel-type spin spirals: $d = \frac{E_{CW} - E_{CCW}}{m}$, in which the constant *m* depends on the wavelength of the spiral. Here, we chose a 4 × 1 supercell as the calculation model (Fig. 5a), which has been successfully implemented in various material systems[70]. The micromagnetic DMI constant *D* can be further calculated as[69] $D = \frac{3\sqrt{2}d}{N_F a^2}$, in which $N_F$ is the number of magnetic layers and *a* is the lattice constant. Since the $C_3$ rotation axis is maintained in Fe$_{3-x}$GaTe$_2$, the DMI is isotropic in plane yielding. The DFT-calculated DMI constant as a function of Fe$_{II}$ displacement is displayed in Fig. 5b. Note that here the Fe$_{II}$ displacements in the upper and lower layers are set to the same value for simplicity. The negative *D* indicates a preferred CW chirality, and its magnitude increases monotonically with the displacement value. Considering that the DMI is essentially derived from the spin-orbit coupling (SOC) effect, the layer-resolved SOC energy difference, $\Delta E_{SOC}$, is calculated to help anatomize the DMI mechanism. When Fe$_{II}$ is not shifted, the $\Delta E_{SOC}$ in each Fe$_{3-x}$GaTe$_2$ monolayer is symmetrically distributed but with opposite signs (Supplementary Fig. 20a), leading to a negligible DMI, which is consistent with our prior analysis. With the emergence of Fe$_{II}$ displacement, the $\Delta E_{SOC}$ distribution becomes asymmetric, and a net $\Delta E_{SOC}$ can be obtained. Note that the

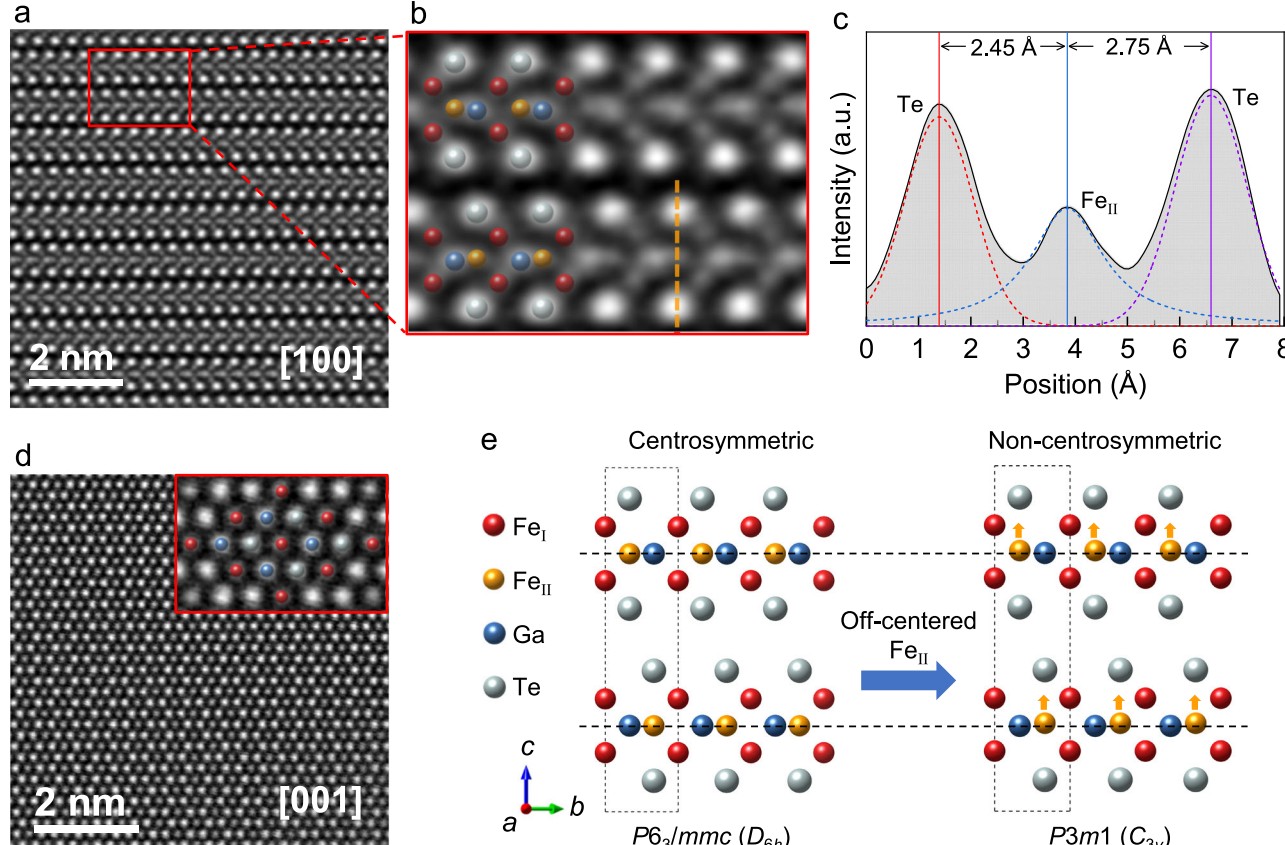

**Fig. 4 | Observation of off-centered Fe$_{II}$ in Fe$_{3−x}$GaTe$_2$. a** HAADF-STEM image taken along the [100] zone axis. **b** Magnified image captured from **a**. **c** Intensity profile of the orange dashed lines in (**b**). The profile is fitted by the Voigt function (dashed curves). The peak positions are indicated by the vertical lines. **d** HAADF-STEM images taken along the [001] zone axis. Inset is a magnified image. **e** Schematic of Fe$_3$GaTe$_2$ crystal structure. The dashed rectangle indicates the unit cell. The off-centered Fe$_{II}$ reduces the crystal symmetry.

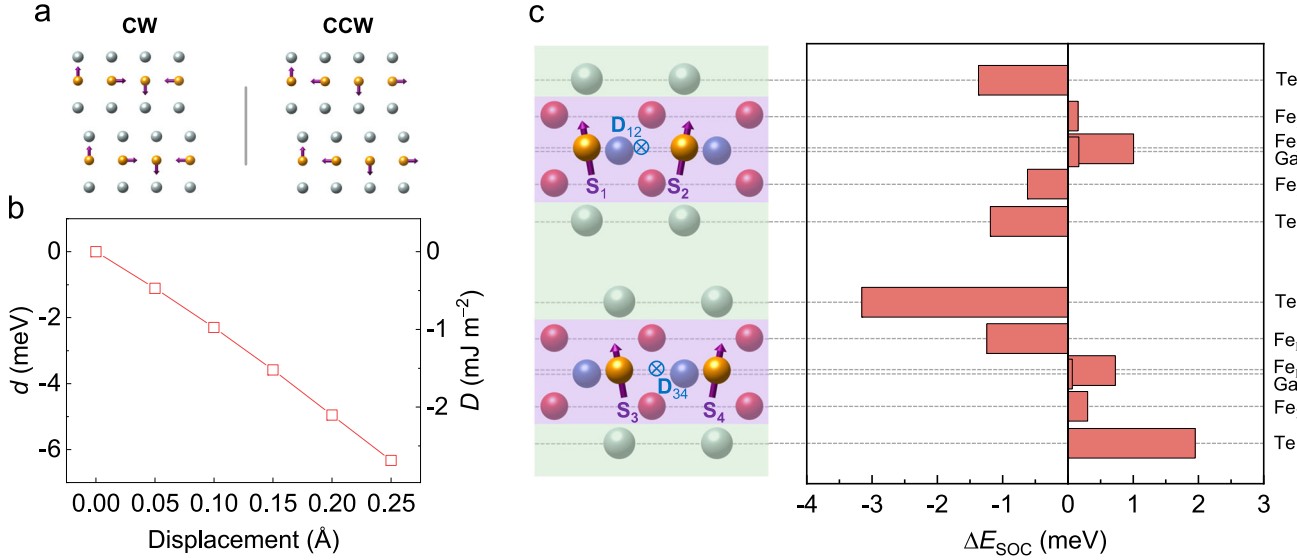

**Fig. 5 | Anatomy of DMI in Fe$_{3−x}$GaTe$_2$. a** Clockwise and counter-clockwise spin spiral models used in the DFT calculation. For clarity, only Te and Fe$_{II}$ atoms are shown. **b** Calculated microscopic and micromagnetic DMI parameters ($d$ and $D$) as a function of Fe$_{II}$ displacement. Note that the Fe$_{II}$ displacements in the upper and lower layers are set to the same value for simplicity. **c** Schematic of spin structure and layer-resolved SOC energy difference $\Delta E_{SOC}$ calculated using the SCXRD-refined crystal structure.

calculated $d$ is negative thus a negative $\Delta E_{SOC}$ corresponds to a positive contribution to the DMI. Large SOC energy difference associated to DMI emerges at the Te sublayers, which is a characteristic feature of the Fert–Lévy model[42,43], suggesting that the DMI mainly stems from the heavy elements through the $Fe_{II}$–Te–$Fe_{II}$ triplet. Figure 5c shows the results for the SCXRD-refined structure. The overall $Fe_{II}$ displacements determined by SCXRD are 0.075 and 0.076 Å in the two layers, leading to a DMI constant of $|D| = 0.50$ mJ m$^{-2}$, which is very close to the experimentally estimated one (i.e. 0.51 mJ m$^{-2}$ at room temperature). Therefore, the first-principles calculations demonstrate that the observed off-centered $Fe_{II}$ can indeed induce an effective DMI via the Fert–Lévy mechanism, making $Fe_{3-x}GaTe_2$ an intrinsic 2D topological ferromagnet.

It is worth mentioning that our first-principles calculations reveal that the stoichiometric $Fe_3GaTe_2$ (without defects) is energetically more favorable to form the centrosymmetric crystal structure rather than the non-centrosymmetric one. Considering the Fe deficiency in our samples, we speculate that the observed $Fe_{II}$ displacement might be caused by defects, such as Fe vacancies and/or interstitial Ga and Te atoms, which can be complex in $Fe_{3-x}GaTe_2$. The small unequal displacements in the two layers might be caused by the nonuniformity of local defect conditions. We would like to leave this interesting topic as an open question for future studies.

Finally, the stabilization mechanism of skyrmions in $Fe_{3-x}GaTe_2$ needs to be clarified. As shown in Fig. 2, the skyrmions in $Fe_{3-x}GaTe_2$ show a significant variation in size and form a disordered lattice. It can be understood that, due to the long-range dipolar interaction, skyrmions repel each other when they are densely packed[71]. The DMI is not strong enough to constrain the skyrmion size, leading to the notable size variation. According to theoretical studies[72], DMI-stabilized skyrmions are insensitive to the external magnetic field in size, while dipolar-stabilized skyrmions are rather sensitive. Supplementary Fig. 16f shows the change of skyrmion size in an increasing positive magnetic field, where the field direction is same as the one in the FC process. One can observe a substantial decrease of skyrmion size with the increasing field, indicating the dominant role of dipolar interaction in the stabilization of skyrmions. The skyrmion evolution in a negative magnetic field is also examined. In Supplementary Fig. 21, skyrmions expand gradually with increasing the negative field and squeeze each other, forming a Voronoi-like network. The apparent field dependence of skyrmion size once again highlights the dominance of dipolar interaction. On the other hand, the Voronoi-like network stems from the topological prevention of annihilation of paired homochiral domain walls[71,73]. Thus, the DMI must exist to maintain the homochirality. Nevertheless, distinct from the nearly equal-sized network cells in chiral helimagnets[74], here they show a significant size variation, suggesting a relatively weaker DMI in $Fe_{3-x}GaTe_2$.

Another effective way to reveal the stabilization mechanism of skyrmions is to investigate the sample-thickness-dependent skyrmion size. Generally, in chiral materials such as B20 magnets, the skyrmion size is determined by the competition between exchange interaction and DMI, and thus almost independent of sample thickness[75]. On the contrary, type-I[76] and type-II bubbles[77] in centrosymmetric materials and even some Néel skyrmions[34,53] and antiskyrmions[78] in non-centrosymmetric materials, where the dipolar interaction prevails, show a strong thickness dependence of size. Supplementary Fig. 22 shows the thickness-dependent skyrmion size in $Fe_{3-x}GaTe_2$ at 295 K at zero magnetic field after an identical FC process. One can observe that the skyrmion size obviously changes with the flake thickness. The non-monotonic variation trend suggests the complicated thickness dependence of dipolar interaction and magnetic anisotropy. Therefore, the magnetic skyrmions in $Fe_{3-x}GaTe_2$ should be primarily dipolar stabilized, and the DMI is essential to guarantee the Néel-type configuration.

In summary, we have demonstrated a new vdW skyrmion-hosting ferromagnet $Fe_{3-x}GaTe_2$. Using L-TEM, isolated Néel-type skyrmions and metastable skyrmion lattices at zero magnetic field are successfully generated by tuning the magnetic field and applying a specific FC procedure, respectively. The metastable chiral skyrmion lattice exhibits remarkable thermal stability even above room temperature, up to 330 K, which is the highest temperature among all known 2D vdW magnets. Furthermore, we have comprehensively investigated the physical origin of the DMI in $Fe_{3-x}GaTe_2$. An unexpected $Fe_{II}$ displacement is experimentally observed, which breaks the spatial inversion symmetry and leads to the $C_{3v}$ point group. First-principles calculations further unveil that the DMI is dominated by the three-site Fert–Lévy mechanism. Our results provide a novel paradigm of 2D topological magnets and reveal the immense potential of $Fe_{3-x}GaTe_2$ for future skyrmion-based applications.

## Methods
### Single crystal growth and characterizations
High-quality $Fe_{3-x}GaTe_2$ single crystals were synthesized using a modified CVT method. The flake thickness was determined by atomic force microscopy (Park NX20). The SCXRD experiment was performed on a Bruker D8 Venture diffractometer with Mo Kα radiation ($\lambda = 0.71073$ Å). The magnetization measurements were carried out in a physical property measurement system with the VSM option (Dyna-Cool, Quantum Design). The $Fe_{3-x}GaTe_2$ Hall-bar device was fabricated by standard electron-beam lithography (Raith eLINE). The electrical transport properties were measured in DynaCool.

### L-TEM and HAADF-STEM experiments
The L-TEM samples were exfoliated from bulk crystals using blue tapes (Ultron Systems). Then they were transferred onto a silicon nitride TEM grid (CleanSiN) using polydimethylsiloxane. The $Fe_{3-x}GaTe_2$/hBN heterostructures were made by stacking another hBN flake on the top following the same transfer procedure. The exfoliation and transfer processes were carried out in a $N_2$-filled glovebox ($O_2 < 0.5$ ppm, $H_2O < 0.1$ ppm). The cross-sectional samples for HAADF-STEM experiments were cut from bulk crystals and thinned by the focused ion-beam milling technique on a TESCAN LYRA3 FIB-SEM system. The SAED and L-TEM images were acquired on an FEI Titan Cs Image (Spectra 300, ThermoFisher Scientific, USA) transmission electron microscope. The HAADF-STEM images were obtained on an FEI Titan Cs Probe (Titan Cubed Themis Z, ThermoFisher Scientific, USA) transmission electron microscope.

### First-principles calculations and simulations
The first-principles calculations were performed using the Vienna ab initio simulation package[79]. The MuMax3 software package[80] was used to perform the micromagnetic simulations of the Néel-type skyrmions, then the MALTS code[81] was adopted to simulate the corresponding L-TEM images.

### Reporting summary
Further information on research design is available in the Nature Portfolio Reporting Summary linked to this article.

## Data availability
The data that support the findings of this study are available from the corresponding author upon reasonable request.

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

## Acknowledgements

This research is supported by the National Research Foundation, Singapore and A*STAR under its Quantum Engineering Programme (NRF2022-QEP2-03-P13), UParis-NUS 2023 award (Inducing magnetic chirality without heavy metals), National Natural Science Foundation of China (nos. 12374011, 91962212, and 51771085); National Key Research and Development Program of China (MOST) (nos. 2022YFC2903504 and 2022YFA1405102).

## Author contributions

C.Z. and H.Y. conceived the idea and designed the experiments. C.Z. synthesized the crystals; C.Z., F.H., S.Z., D.Y. and Y.L. did the basic characterizations; Z.J. and W.H. operated the L-TEM and HAADF-STEM experiments under the guidance of J.Z. and Y.P.; J.J. carried out the first-principles calculations under the guidance of Hx.Y.; C.Z., J.J. and H.Y. analyzed the data and wrote the manuscript. All authors contributed to the discussion of the results and the improvement of the manuscript. H.Y. led the project.

## Competing interests

The authors declare no competing interests.
