## [Peer Review File · Nature Communications]

Reviewers' Comments:

Reviewer #1:

Remarks to the Author:

The work by C. Zhang et al reports an investigation into skyrmion formation in the van der Waals magnet Fe₃GaTe₂, utilising a commendably large range of characterisation techniques, including magnetometry, transport, and Lorentz transmission electron microscopy, as well as DFT calculations. The main claims of the work are that the observation of monochiral Neel-type skyrmions must indicate the presence of the Dzyaloshinskii-Moriya interaction (DMI). The authors motivate this by various calculations and the aforementioned DFT.

Overall, I have several concerns regarding the LTEM experiments, and some of the calculations. Interpretation of LTEM data can always be a challenge, and here the author's use of very large defocus values when imaging prohibits quantitative analysis of the LTEM images. Moreover, the experimental calculation of the DMI may require some more careful considerations. From the images of the bubble-like skyrmions, I believe that the dipolar interaction may still be the dominant stabilisation energy term of the skyrmions. However, the observation of Neel-type domain walls requires explanation.

Nevertheless, of all the mechanisms proposed to allow for an interfacial-like DMI in van der Waals systems (of which there are now numerous in the literature), the author's HAADF-STEM images in Fig. 4 may be the most convincing I have seen. If this is correct, and the results truly show the broken symmetry of the Fe (II) site position, this may indeed provide the DMI required for Neel-type domain wall formation. This finding also has important implications for other 2D magnets: this study motivates the question of whether similar phenomenon may be occurring in the related Neel-type skyrmion host, Fe₃GeTe₂. However, at the moment due to the lower resolution and small size of the corresponding plots, it's difficult for me to confirm this result.

Overall, given the authors have demonstrated skyrmion formation in a room temperature van der Waals magnet, and have possibly found an explanation for observations Neel-type skyrmions in these systems, I am leaning towards recommending the work for publication, if the authors can address the comments and concerns I have, listed below. Most importantly, it must be established beyond doubt that there really is an inversion breaking in the STEM data, in order to support the authors' conclusions.

1) The authors state "the skyrmion lattice shows enhanced topological stability" compared to isolated skyrmions. I believe this is not true: the topology of each skyrmion is defined individually and distinctly, and their clustering into a lattice shouldn't have an impact.

2) The authors state "Generally, the skyrmion lattice phase is confined to a small H-T region just below T_c". I believe this is only true for the original helimagnet skyrmions in their bulk form (B20 materials etc). Thin plates of the original bulk skyrmion materials, and more recently investigated bubble-like materials all show a large, expansion skyrmion phase.

3) I have several comments about the LTEM measurements:

a) My biggest concern is the authors' use of a very large defocus value of 3mm. Defocusing of the electron beam is of course required to achieve magnetic contrast. However, a large defocus such as this has the possibility to blur the underlying magnetic structures considerably. Indeed, this may be the reason that the authors observe the curious type-II bubble-like contrast in their largest skyrmions. I would like to see the authors perform a defocus series between +/- 3mm, at both +15, 0, and -15 degrees. This will be very valuable to confirm that the observed textures are indeed Neel-type, and may give more detail about their internal structure (nature of this additional contrast).

b) The authors seem to indicate that their observed spin textures are primarily stabilised by interfacial DMI. However, I believe that the large variation in skyrmion size and the disordered nature of the lattice may indicate that the bubbles are primarily dipolar stabilised. I would urge the

authors to consider the work by F. Büttner et al [F. Büttner, et al. Sci. Reps. 8, 4464 (2018)], which shows that there is more of a spectrum of skyrmion/bubble stability. Specifically, it's possible for skyrmions to be primarily dipolar stabilised, but have their domain wall chirality and helicity determined by the presence of DMI (Neel-type in this case). Another example is the antiskyrmion materials, where the objects are likely mostly dipolar-stabilised, but the antiskyrmion-like domain walls are defined by the anisotropic DMI.

c) Along these lines, as suggested by F. Büttner's paper, investigating the field dependent size of the skyrmions gives a solid indication of the primary stabilisation mechanism: dipolar stabilised skyrmions will change size dramatically with field, whereas DMI-stabilised skyrmions do not. Specifically, nucleating skyrmions with a positive applied FC, and then applying negative fields should be effective. Another key observation would be the size of the spin textures in samples of different thickness. The authors have investigated two thicknesses here, is there an appreciable difference between the spin texture size at the same T and H? The authors may wish to consider this, or perhaps they already have such data available.

d) I noticed in several of the supplementary images, the authors have observed skyrmionium states (closed loops of domain walls), but make no mention of them. For example, Fig. S12 shows several, and Fig. S6d, e. These states were recently observed in Fe₃GeTe₂ [L. Powalla et al. Adv. Mater. 35, 2208930 (2023)], and the authors may wish to comment on their formation in this new material in the present work.

4) The authors compare their work to other van der Waals materials. At the bottom of page 6, it might be pertinent to compare to other skyrmion systems where skyrmions may form at room temperature (the CoZnMn alloys, and aforementioned antiskyrmion compounds).

5) In Fig. 3, it is difficult to know from which work each data point has come from. The authors should probably better label this.

6) I have several issues with the calculation for an experimental value of the DMI, which I think should be directly indicated to be an estimation, rather than a precise and accurate value. The authors make use of several quantities to calculate this value, largely detailed in Supplementary Note 2. The first is the exchange stiffness A, which the authors acquire from their magnetometry data (via B and D_{spin}). The second is the effective anisotropy K_{eff}, which they acquire from the magnetometry measurements along both the out-of-plane and in-plane directions. The third is the estimated domain wall energy, acquired from the measurement of the average domain wall width in the LTEM data, and the saturation magnetisation.

a) Given the large number of parameters (not to mention the $\gamma=1$ and $\beta=0.31$ assumptions), I find it difficult to believe the accuracy and precision of the authors' DMI value. Have the authors tried to perform some error analysis, to consider how large the error in their DMI value could be, given the error in each of these other quantities?

b) Why do the authors only consider 5 data points in their Fig. S9, while they have clearly measured M_vH at more temperatures (from the Fig. 1 data)?

c) I have further specific issues with the K_{eff} value. Firstly, the M_vH loops clearly show hysteresis in Fig. 1b (is there any for the IP direction?). How is the hysteresis considered when calculating the K_{eff} value? More importantly, the value of K_{eff} has been estimated from the bulk crystal. However, it is well known that shape and surface anisotropies are also relevant. Is this K_{eff} value from the bulk crystal really representative of the K_{eff} of the exfoliated flakes? Could it be possible to acquire a more accurate value of K_{eff} directly for flakes using transport measurements (I suppose this would require performing IP field transport measurements).

7) The HAADF-STEM images in Fig. 4 are perhaps the most crucial results for the authors' arguments. Once again, I have several/comments questions about these results

a) Due to the lower resolution of my referee files in combination with the small size of panel f, it is

difficult to see, and therefore believe, this Fe site displacement result. I would like to see the authors greatly enlarge panel 4f, and add more detail. For example, could they fit the three peaks with Gaussian/Lorentz functions, to more accurately determine the central position of each atom, and indicate these values much more clearly on the plot? From my own analysis of the poor resolution image using ImageJ, the displacement must be quite small, so careful analysis must be clearly shown.

b) I was wondering if the authors have investigated other portions of the crystal? I am curious to see if the same inversion breaking direction is seen throughout the crystal. Do the authors have any speculation as to how/why the crystal would uniformly pick one breaking of the inversion symmetry?

c) Is it possible to use the x-ray diffraction data to perform a structural refinement, and corroborate the broken inversion symmetry of the Fe(ii) atom site? I imagine this change in symmetry should result in the appearance of small diffraction peaks, which may be forbidden in the centrosymmetric crystal structure.

d) I am wondering if the authors utilised their DFT simulations to check the energy difference between the centrosymmetric and non-centrosymmetric crystal structures (forgetting about magnetism for the moment). Is it energetically favourable to form this structure, or is the energy cost relatively small? Any clues about why the symmetry would break in this way?

e) The authors may wish to compare their work to another Neel-type skyrmion host, also a Polar magnet: GaV4S8 [I. Kesmarki et al. Nat. Mater. 14, 1116-1122 (2015)].

8) Excluding the presence of oxidised layers is a crucial point when comparing to other works in the literature. Did the authors perform any measurements to characterise the oxidisation of their samples (even the best glovebox atmosphere may lead to some oxidisation)?

9) I'm not so familiar with the method, but does the DMI calculated from the DFT depend on the assumed spin structure (in the present case, the authors have utilised a 90 degree neighbouring spin rotation between cells). I assume that this is not the case?

Overall, I believe that the observation of Neel-type skyrmions, together with the broken inversion symmetry of the crystal structure, are already vital findings for the community, and this is already enough to warrant publication. It is not necessary to try to accurately and precisely determine the DMI value in the sample (which is notoriously difficult). Although an estimation is welcome, as long as it is clearly an estimation. For me, the most important consideration before acceptance is the careful and clear confirmation of the broken inversion symmetry in the STEM images.

Reviewer #2:

Remarks to the Author:

In this work by C. Zhang et. al.. the authors report on the observation of Néel type skyrmions in Fe₃-xGaTe₂. They analyze the magnetic properties of the bulk material to determine above room temperature Curie point, as well as magnetocrystalline anisotropy along the c-axis. The LTEM data is presented which shows clearly the domains formed are of Néel type. Following a FC protocol, they are able to stabilize the skyrmion lattice which is a meta-stable phase and demonstrated by heating the sample close to Curie temperature. The origin of interfacial DMI is then presented which is shown to be due to the off-centering of the Fe(II) sites. This is confirmed through high resolution STEM imaging as well as DFT. Overall this work is very well done, and comprehensively shows the formation, and origin of Néel type skyrmions in this material at RT. The paper is recommended for publication in Nat. Comm. with some revisions as noted below.

1) Can the authors comment about the reason for the change in the slope of the magnetization curve? Is it caused by defects impeding the movement of the domains or is there a secondary phase contribution?

2) The procedure to obtain skyrmion lattice is mis-represented by calling it zero-field skyrmions. This method of generating skyrmions has been pretty well established, and as the authors mention, they are nucleated only during FC process where a field is necessary. After the removal of the field at room temperature, the skyrmions still exist because they form a metastable state. This should not be referred to as zero-field skyrmions.

3) Can the authors comment on the origin of the Fe(II) ion shift i.e. the origin for it? Is it due to the off-stoichiometry of the compound i.e. being iron deficient, or is there another reason. And what could be ways of controlling this behavior?

Minor:

Definition of in-plane and out-of-plane is incorrect – $H // c$, and $H \perp c$ on last line page 3.

Reviewer #3:

Remarks to the Author:

This work by C. Zhang et. al. reports on the realization of room-temperature chiral skyrmion lattice in a van der Waals ferromagnet $\text{Fe}_{3-x}\text{GaTe}_2$. Given that the $\text{Fe}_{3-x}\text{GaTe}_2$ is known to be a centrosymmetric structure, the observed chiral skyrmion lattice can be a surprising. The authors attribute the chiral skyrmion to unexpected displacement of Fe atoms, leading to the Dzyaloshinskii-Moriya interaction (DMI). While their results are interesting, and the claims are well supported by robust experimental and theoretical evidences, I am a bit negative to the publication of this work in Nature Communications for the following reasons below.

First, room-temperature chiral Neel skyrmions in 2D van der Waals material is already reported in 2022 [ref. 35]. While the present work emphasizes that their skyrmion is stabilized above-room temperature, I think this is not surprising progress because anyhow room-temperature skyrmion means that the skyrmion can be observed above room-temperature. This is just matter of how wide the temperature range is.

Second, the controllability of the DMI reported in the manuscript is quite skeptical. In the introduction, the authors listed up several previous results which report ways to induce inversion symmetry breaking and thus the DMI, and the authors pointed out that the previous reports still have limitations because these approaches are challenging to control. However, I am not sure the DMI in proposed this work is superior in terms of controllability, meaning that this work still has the same limitation.

Third, whereas the origin of the DMI in the $\text{Fe}_{3-x}\text{GaTe}_2$ is firmly confirmed, the manuscript lacks information regarding why the FeII atoms are displaced and how the direction of the displacement is determined in the crystal are missing in the manuscript. I think these are also important factors if one to develop skyrmion-devices using 2D materials with high controllability.

For the reasons above, I am hesitating to recommend the publication of this manuscript in Nature Communications.

Response to Reviewer #1

The work by C. Zhang et al reports an investigation into skyrmion formation in the van der Waals magnet Fe_3GaTe_2 , utilising a commendably large range of characterisation techniques, including magnetometry, transport, and Lorentz transmission electron microscopy, as well as DFT calculations. The main claims of the work are that the observation of monochiral Neel-type skyrmions must indicate the presence of the Dzyaloshinskii-Moriya interaction (DMI). The authors motivate this by various calculations and the aforementioned DFT.

Overall, I have several concerns regarding the LTEM experiments, and some of the calculations. Interpretation of LTEM data can always be a challenge, and here the author's use of very large defocus values when imaging prohibits quantitative analysis of the LTEM images. Moreover, the experimental calculation of the DMI may require some more careful considerations. From the images of the bubblelike skyrmions, I believe that the dipolar interaction may still be the dominant stabilisation energy term of the skyrmions. However, the observation of Neel-type domain walls requires explanation.

Nevertheless, of all the mechanisms proposed to allow for an interfacial-like DMI in van der Waals systems (of which there are now numerous in the literature), the author's HAADF-STEM images in Fig. 4 may be the most convincing I have seen. If this is correct, and the results truly show the broken symmetry of the Fe (II) site position, this may indeed provide the DMI required for Neel-type domain wall formation. This finding also has important implications for other 2D magnets: this study motivates the question of whether similar phenomenon may be occurring in the related Neel-type skyrmion host, Fe_3GeTe_2 . However, at the moment due to the lower resolution and small size of the corresponding plots, it's difficult for me to confirm this result.

Overall, given the authors have demonstrated skyrmion formation in a room temperature van der Waals magnet, and have possibly found an explanation for observations Neel-type skyrmions in these systems, I am leaning towards recommending the work for publication, if the authors can address the comments and concerns I have, listed below. Most importantly, it must be established beyond doubt that there really is an inversion breaking in the STEM data, in order to support the authors' conclusions.

Reply: We deeply appreciate the referee's careful review and very valuable comments. The referee mainly concerns the L-TEM and HAADF-STEM data, which are key evidence for the observation of Néel-type skyrmions and inversion symmetry breaking. Another concern is the estimation of DMI using experimental parameters. In light of these concerns, we have performed additional experiments to answer the comments, which will be shown in the point-by-point responses. All the modifications based on the referees' comments are highlighted in red in the revised manuscript.

1) The authors state “the skyrmion lattice shows enhanced topological stability” compared to isolated skyrmions. I believe this is not true: the topology of each skyrmion is defined individually and distinctly, and their clustering into a lattice shouldn’t have an impact.

Reply: We thank the referee for this comment. In the paper we cite [*Nano Lett.* 18, 7362-7371 (2018)], the chiral skyrmion bubble lattice is created using ultrafast laser pulse under a positive magnetic field. The authors find that the skyrmion bubble lattice can survive under a stronger negative perturbative magnetic field than the isolated skyrmions, implying the role of the bubble–bubble interaction in the skyrmion lattice. In this perspective, they claim that the skyrmion bubble lattice leads to an enhanced topological stability as compared to isolated skyrmions, Nevertheless, under the positive perturbative magnetic field, no considerable enhanced topological stability is observed. Therefore, after reconsideration, we think the enhanced topological stability in skyrmion lattice compared to isolated skyrmions might be valid only in very specific cases. Thus, we delete this statement in the revised manuscript.

2) The authors state “Generally, the skyrmion lattice phase is confined to a small H - T region just below T_c ”. I believe this is only true for the original helimagnet skyrmions in their bulk form (B20 materials etc). Thin plates of the original bulk skyrmion materials, and more recently investigated bubble-like materials all show a large, expansion skyrmion phase.

Reply: As the referee pointed out, for the original helimagnets (B20 materials, etc.) in their bulk form, the skyrmion lattice usually appears in a small H - T region just below T_c . It is also true that recent studies report the wide-temperature-range skyrmion phase in some bulk skyrmion materials [*Phys. Rev. B* 95, 144403 (2017); *ACS Appl. Mater. Interfaces* 12, 24125–24132 (2020); *Commun. Phys.* 5, 189 (2022); *ACS Nano* 16, 13911-13918 (2022)], and they are primarily centrosymmetric and thus display Bloch skyrmion bubbles. Whereas in the newly discovered van der Waals Néel-skyrmion-hosting materials such as Fe_3GeTe_2 and $(\text{Fe}_{0.5}\text{Co}_{0.5})_5\text{GeTe}_2$, the equilibrium skyrmion phase is also confined to a small H - T region just below T_c (see Figs. R1 and R2), which is very similar to the original helimagnets.

Fig. R1. History and thickness dependent magnetic phase diagrams. Magnetic phase diagrams determined by imaging the Fe_3GeTe_2 flake as a function of temperature and applied field for three different thickness regions: 60 nm (a–c), 50 nm (d–f) and 35 nm (g–i). Phase diagrams were determined by following the zero field cooling (ZFC) (a, d, g), field sweep (FS) (b, e, h), or field cooling (FC) at 15 mT (c, f, i) measurement procedures. The measurement paths are indicated by the red arrows in a–c. The existence of the stripe domain (SD, orange), skyrmion (Sk^\pm , green) and uniformly magnetised (UM^+ , yellow, UM^- , purple) states are indicated by the coloured regions. Reproduced from [Nat. Commun. 13, 3035 (2022)].

Fig. R2. History Phase diagram of $(\text{Fe}_{0.5}\text{Co}_{0.5})_5\text{GeTe}_2$ as a function of temperature and magnetic field obtained in 136-nm-thick nanoflakes. The blue, green, yellow, and purple areas represent skyrmion lattice (SkL), stripe, ferromagnetic (FM), and paramagnetic phases (PM), respectively. Reproduced from [Sci. Adv. 8, eabm7103 (2022)].

Compared to the skyrmion-bubble-hosting materials, we think $\text{Fe}_{3-x}\text{GaTe}_2$ should be more similar to Fe_3GeTe_2 and $(\text{Fe}_{0.5}\text{Co}_{0.5})_5\text{GeTe}_2$. Therefore, we rephrase the corresponding sentences on page 6 as follows:

“...Generally, in many bulk helimagnets and vdW ferromagnetic thin flakes such as Fe_3GeTe_2 and $(\text{Fe}_{0.5}\text{Co}_{0.5})_5\text{GeTe}_2$, the equilibrium skyrmion lattice phase is confined to a small H - T region just below T_c ^{4,17,26}. However, once the sample is cooled from the paramagnetic state with a suitable magnetic field and passes through this phase pocket, the skyrmion lattice emerges and it can persist even outside of this region due to the topological stability...”

3) I have several comments about the LTEM measurements:

a) My biggest concern is the authors' use of a very large defocus value of 3mm. Defocusing of the electron beam is of course required to achieve magnetic contrast. However, a large defocus such as this has the possibility to blur the underlying magnetic structures considerably. Indeed, this may be the reason that the authors observe the curious type-II bubble-like contrast in their largest skyrmions. I would like to see the authors perform a defocus series between +/- 3mm, at both +15, 0, and -15 degrees. This will be very valuable to confirm that the observed textures are indeed Néel-type, and may give more detail about their internal structure (nature of this additional contrast).

Reply: To answer this comment, we perform additional L-TEM experiments on the defocus- and tilt-angle-dependent imaging of Néel-type skyrmions in $\text{Fe}_{3-x}\text{GaTe}_2$. The results are shown in Supplementary Fig. 8 (Fig. R3 in this response letter). One can observe that the magnetic contrast is gradually enhanced with the increase of defocus value at a fixed tilt angle. Without tilting, no magnetic contrast can be observed. Thus, these magnetic domains are Néel-type.

Under a large defocus (≥ 1 mm), some big-sized skyrmions show additional contrast compared to the common dark-bright spots of Néel-type skyrmions. In fact, this phenomenon is also observed in some other studies [*Adv. Mater.* 32, 1904327 (2020); *Nat. Commun.* 13, 3965 (2022)], as shown in Figs. R4 and R5. Further, reducing the defocus to ± 0.5 mm can help suppress the additional contrast, as shown in Fig. R3. However, the magnetic contrast is also greatly compromised.

Fig. R3. Supplementary Fig. 8 | Defocus- and tilt-angle-dependent L-TEM images of skyrmion lattice. Before imaging, the sample is FC from 370 to 295 K under an out-of-plane magnetic field of 360 Oe, and subsequently the field is removed.

Fig. R4. **a,b** Under-focused (1.2 mm) L-TEM images of Néel skyrmions recorded at 0 T **(a)** and 0.38 T **(b)** after field cooling in a perpendicular field of 0.096 T in PtMnGa. **c** Simulated L-TEM contrast for skyrmions of different diameters versus tilt angle in degrees along the x - and y -axis. Reproduced from [*Adv. Mater.* 32, 1904327 (2020)].

Fig. R5. Evolution of Néel-type textures in a thin lamella of $\text{Cr}_{1+\delta}\text{Te}_2$ as a function of magnetic field at 100 K. **a–h** L-TEM images of Néel-type textures collected at different magnetic fields by tilting the lamella 7° away from the [0111] direction. These images are acquired at a defocus value of -1.2 mm. Reproduced from [*Nat. Commun.* 13, 3965 (2022)].

Therefore, the additional contrast in some Néel skyrmions might be caused by the relatively large skyrmion size and the large defocus we use to enhance the magnetic contrast. We thank the referee for this valuable comment, and we add the following discussion in the revised manuscript on page 7:

“Similar Néel skyrmion images with additional contrast inside are also observed in some other ferromagnets such as $\text{Cr}_{1.3}\text{Te}_2$ ³⁴ and PtMnGa ⁵³. In Supplementary Fig. 8, we perform more L-TEM experiments on the defocus- and tilt-angle-dependent imaging of Néel-type skyrmions in $\text{Fe}_{3-x}\text{GaTe}_2$. One can observe that the magnetic contrast is gradually enhanced with the increase of defocus at a fixed tilt angle. The additional contrast inside the skyrmion is greatly suppressed when the defocus decreases to ± 0.5 mm, accompanied by a significant compromise of magnetic contrast at the same time. Therefore, the additional contrast in some Néel skyrmions should be caused by the relatively large skyrmion size and the large defocus that is used to enhance the magnetic contrast.”

b) The authors seem to indicate that their observed spin textures are primarily stabilised by interfacial DMI. However, I believe that the large variation in skyrmion size and the disordered nature of the lattice may indicate that the bubbles are primarily dipolar stabilised. I would urge the authors to consider the work by F. Büttner et al [F. Büttner, et al. *Sci. Reps.* 8, 4464 (2018)], which shows that there is more of a spectrum of skyrmion/bubble stability. Specifically, it's possible for skyrmions to be primarily dipolar stabilised, but have their domain wall chirality and helicity determined by the presence of DMI (Neel type in this case). Another example is the antiskyrmion materials, where the objects are likely mostly dipolar-stabilised, but the antiskyrmion-like domain walls are defined by the anisotropic DMI.

c) Along these lines, as suggested by F. Büttner's paper, investigating the field dependent size of the skyrmions gives a solid indication of the primary stabilisation mechanism: dipolar stabilised skyrmions will change size dramatically with field, whereas DMI-stabilised skyrmions do not. Specifically, nucleating skyrmions with a positive applied FC, and then applying negative fields should be effective. Another key observation would be the size of the spin textures in samples of different thickness. The authors have investigated two thicknesses here, is there an appreciable difference between the spin texture size at the same T and H? The authors may wish to consider this, or perhaps they already have such data available.

Reply: We would like to combine these two comments together because they are both related to the stabilization mechanism of skyrmions in $\text{Fe}_{3-x}\text{GaTe}_2$. We agree with the referee that the main stabilization mechanism should be the interplay of dipolar interaction and uniaxial magnetic anisotropy, and the DMI is necessary to guarantee the Néel-type spin configuration. Generally, in conventional helimagnets, nearly equal-sized skyrmions form an ordered lattice [*Nature* 465, 901-904 (2010); *Nat. Mater.* 10, 106-109 (2011)]. However, in $\text{Fe}_{3-x}\text{GaTe}_2$ the skyrmions show a significant variation in size and form a disordered array. It can be understood that, due to the long-range dipolar interaction, skyrmions repel each other when they are densely packed [*Nano Lett.*

18, 7362-7371 (2018)]. While the DMI is not strong enough to constrain the skyrmion size as it does in chiral helimagnets. Following the referee's suggestions, we also perform more L-TEM experiments to reveal the skyrmion stabilization mechanism.

First, we extract the average skyrmion size in an increasing positive magnetic field and plot them in Supplementary Fig. 16 (Fig. R6 in this response letter). According to theoretical studies [F. Büttner, et al. *Sci. Reps.* **8**, 4464 (2018)], DMI-stabilized skyrmions are insensitive to the external magnetic field in size, while dipolar-stabilized skyrmions are rather sensitive. In Fig. R6f, one can observe a substantial decrease of skyrmion size with the increasing field, indicating the dominant role of dipolar interaction in the stabilization of skyrmions.

Second, we also examine the skyrmion evolution in a negative magnetic field as the referee suggested. In Supplementary Fig. 19 (Fig. R7 in this response letter), the apparent field dependence of skyrmion size once again highlights the dominance of dipolar interaction.

Third, we further investigate the sample-thickness-dependent skyrmion size in $\text{Fe}_{3-x}\text{GaTe}_2$. Generally, in chiral materials such as B20 magnets [*Nat. Mater.* **10**, 106-109 (2011)], the skyrmion size is determined by the competition between exchange interaction and DMI and thus almost independent of sample thickness. While in dipolar-stabilized systems [*Nat. Commun.* **5**, 3198 (2014); *Adv. Mater.* **32**, e2002043 (2020); *Nat. Commun.* **13**, 3965 (2022); *Adv. Mater.* **32**, 1904327 (2020); *Nat. Commun.* **13**, 3035 (2022)], the spin texture size is determined by the interplay of dipolar interaction and uniaxial magnetic anisotropy, showing a strong thickness dependence. Supplementary Figure 20 (Fig. R8 in this response letter) shows the thickness-dependent skyrmion size in $\text{Fe}_{3-x}\text{GaTe}_2$ at 295 K at zero magnetic field after an identical FC process. One can observe that the skyrmion size obviously changes with the flake thickness. The non-monotonic variation trend may indicate the complicated thickness dependence of dipolar interaction and magnetic anisotropy. Therefore, the magnetic skyrmions in $\text{Fe}_{3-x}\text{GaTe}_2$ should be primarily dipolar stabilized, and the DMI is essential to guarantee the Néel-type configuration.

Fig. R6. Supplementary Fig. 16 | Skyrmion size change in a positive magnetic field. Before imaging, the sample is FC from 370 to 300 K under an out-of-plane magnetic field of 360 Oe, and subsequently the field is removed. Then the field gradually increases from a to e. In f, d_{sk} represents the diameter of the skyrmion. The defocus value is -3 mm and the tilt angle is -13° .

Fig. R7. Supplementary Fig. 19 | Skyrmion evolution in a negative magnetic field. Before imaging, the sample is FC from 370 to 295 K under an out-of-plane magnetic field of 360 Oe, and subsequently the field is removed. Then an opposite magnetic field gradually increases from 0 to -1.88 kOe. The defocus value is -2 mm and the tilt angle is 15° .

Fig. R8. Supplementary Fig. 20 | Sample thickness dependence of skyrmion size. Before imaging, the sample is FC from 370 to 295 K under an out-of-plane magnetic field of 360 Oe, and subsequently the field is removed. The defocus value is -3 mm and the tilt angle is 11° . d_{sk} represents the diameter of the skyrmion.

Here we would like to thank the referee for the constructive comments that further improve our manuscript. Accordingly, we add the following content on pages 13 and 14:

“Finally, the stabilization mechanism of skyrmions in $\text{Fe}_{3-x}\text{GaTe}_2$ needs to be clarified. As shown in Fig. 2, the skyrmions in $\text{Fe}_{3-x}\text{GaTe}_2$ show a significant variation in size and form a disordered lattice. It can be understood that, due to the long-range dipolar interaction, skyrmions repel each other when they are densely packed⁷⁰. The DMI is not strong enough to constrain the skyrmion size, leading to the notable size variation. According to theoretical studies⁷¹, DMI-stabilized skyrmions are insensitive to the external magnetic field in size, while dipolar-stabilized skyrmions are rather sensitive. Supplementary Figure 16f shows the change of skyrmion sizes in an increasing positive magnetic field, where the field direction is same as the one in the FC process. One can

observe a substantial decrease of skyrmion size with the increasing field, indicating the dominant role of dipolar interaction in the stabilization of skyrmions. The skyrmion evolution in a negative magnetic field is also examined. In Supplementary Fig. 19, skyrmions expand gradually with increasing the negative field and squeeze each other, forming a Voronoi-like network. The apparent field dependence of skyrmion size once again highlights the dominance of dipolar interaction. On the other hand, the Voronoi-like network stems from the topological prevention of annihilation of paired homochiral domain walls^{70,72}. Thus, the DMI must exist to maintain the homochirality. Nevertheless, distinct from the nearly equal-sized network cells in chiral helimagnets⁷³, here they show a significant size variation, suggesting a relatively weaker DMI in $\text{Fe}_{3-x}\text{GaTe}_2$.

Another effective way to reveal the stabilization mechanism of skyrmions is to investigate the sample-thickness-dependent skyrmion size. Generally, in chiral materials such as B20 magnets, the skyrmion size is determined by the competition between exchange interaction and DMI, and thus almost independent of sample thickness⁷⁴. On the contrary, type-I⁷⁵ and type-II bubbles⁷⁶ in centrosymmetric materials and even some Néel skyrmions^{34,53} and antiskyrmions⁷⁷ in non-centrosymmetric materials, where the dipolar interaction prevails, show a strong thickness dependence of size. Supplementary Figure 20 shows the thickness-dependent skyrmion size in $\text{Fe}_{3-x}\text{GaTe}_2$ at 295 K at zero magnetic field after an identical FC process. One can observe that the skyrmion size obviously changes with the flake thickness. The non-monotonic variation trend suggests the complicated thickness dependence of dipolar interaction and magnetic anisotropy. Therefore, the magnetic skyrmions in $\text{Fe}_{3-x}\text{GaTe}_2$ should be primarily dipolar stabilized, and the DMI is essential to guarantee the Néel-type configuration.”

d) I noticed in several of the supplementary images, the authors have observed skyrmionium states (closed loops of domain walls), but make no mention of them. For example, Fig. S12 shows several, and Fig. S6d, e. These states were recently observed in Fe_3GeTe_2 [L. Powalla et al. *Adv. Mater.* 35, 2208930 (2023)], and the authors may wish to comment on their formation in this new material in the present work.

Reply: Thanks a lot for this suggestion. The referee is correct, there are skyrmioniums in some images. In light of this comment, we point out them by green arrows in Supplementary Figs. 6 and 14.

Fig. R9. Supplementary Fig. 6 | Field-induced room-temperature magnetic skyrmions in $\text{Fe}_{3-x}\text{GaTe}_2$. a–h Evolution of magnetic domains in a 179 nm-thick $\text{Fe}_{3-x}\text{GaTe}_2$ flake under an out-of-plane magnetic field. The sample is ZFC from 370 to 295 K before imaging. The defocus value is -3 mm and the tilt angle α is 11° . Several Néel-type magnetic skyrmions and skyrmioniums are pointed out by the red and green arrows, respectively.

Fig. R10. Supplementary Fig. 14 | Field-dependent magnetic domain evolution in hBN-protected $\text{Fe}_{3-x}\text{GaTe}_2$. The sample is ZFC from 370 to 320 K before imaging. The defocus value is -3 mm and the tilt angle is 13° . Several Néel-type magnetic skyrmions and skyrmioniums are pointed out by the red and green arrows, respectively.

Skyrmioniums are also observed in other van der Waals materials $\text{Fe}_{3-x}\text{GeTe}_2$ [*Adv. Mater.* **35**, 2208930 (2023)] and $\text{Cr}_2\text{Ge}_2\text{Te}_6$ [*Phys. Rev. B* **108**, 214417 (2023)]. Accordingly, in the revised manuscript, we add a brief discussion on pages 5 and 6:

“More interestingly, we also observe the magnetic skyrmionium state in $\text{Fe}_{3-x}\text{GaTe}_2$, as pointed out by the green arrows in Supplementary Fig. 6. The skyrmionium consists of a uniformly magnetized core region encircled by a 360° domain wall loop⁴⁹. Unlike skyrmions that have a unitary topological charge, skyrmioniums show zero topological charge⁴⁹. As skyrmioniums do not exhibit the skyrmion Hall effect when driven by a spin polarized current, they are considered promising candidates for racetrack memory applications⁵⁰. Moreover, the skyrmionium is observed to transform into a skyrmion when the magnetic field increases from 1.47 to 1.59 kOe (Supplementary Fig. 6f). Similar phenomena are also observed in $\text{Fe}_{3-x}\text{GeTe}_2$ ⁵¹ and $\text{Cr}_2\text{Ge}_2\text{Te}_6$ ⁵² at low temperatures. The diversity of topological spin textures and their phase transitions reveal that $\text{Fe}_{3-x}\text{GaTe}_2$ is a rich platform for room-temperature skyrmionics.”

4) The authors compare their work to other van der Waals materials. At the bottom of page 6, it might be pertinent to compare to other skyrmion systems where skyrmions may form at room temperature (the CoZnMn alloys, and aforementioned antiskyrmion compounds).

Reply: We thank the referee for this suggestion. Accordingly, we add discussion on page 8:

“ β -Mn-type Co-Zn-Mn alloy and B20-type $\text{Co}_{1.043}\text{Si}_{0.957}$ are two rare examples that can host chiral skyrmions at room temperature but in Bloch-type⁵⁷⁻⁵⁹. While in some antiskyrmion compounds, such as $\text{Mn}_{1.4}\text{Pt}_{0.9}\text{Pd}_{0.1}\text{Sn}$ and $\text{Fe}_{1.9}\text{Ni}_{0.9}\text{Pd}_{0.2}\text{P}$, the antiskyrmion phase can reach as high as 400 K^{60,61}. It is worth mentioning that, in current-driven skyrmion motion, Joule heating is inevitable because a large current density is required to overcome the pinning and achieve high-speed motion, which can cause a temperature rise of several tens of kelvin⁶². Therefore, skyrmion materials with a skyrmion phase temperature just reaching room temperature may not be able to function as practical devices at room temperature. The observation of above-room-temperature high-density chiral skyrmion lattice at zero magnetic field in $\text{Fe}_{3-x}\text{GaTe}_2$ opens promising perspectives for developing novel spintronic devices based on 2D magnets.”

5) In Fig. 3, it is difficult to know from which work each data point has come from. The authors should probably better label this.

Reply: Thanks for this suggestion. In the new Fig. 3 (Fig. R11 in this rebuttal letter), we label every data point with its reference:

Fig. R11. Fig. 3 | Above-room-temperature skyrmion lattice at zero magnetic field in $\text{Fe}_{3-x}\text{GaTe}_2$. **a** Temperature dependence of skyrmion lattice in a 179 nm-thick $\text{Fe}_{3-x}\text{GaTe}_2$ flake. The sample is initially FC from 370 to 295 K under an out-of-plane magnetic field of 360 Oe. The defocus value is -3 nm. **b** Comparison of skyrmion lattices at zero magnetic field in various vdW and quasi-vdW magnets^{15,17,22,23,29,34}. The markers # and * indicate Bloch-type skyrmion bubbles and Néel-type skyrmions, respectively.

6) I have several issues with the calculation for an experimental value of the DMI, which I think should be directly indicated to be an estimation, rather than a precise and accurate value. The authors make use of several quantities to calculate this value, largely detailed in Supplementary Note 2. The first is the exchange stiffness A , which the authors acquire from their magnetometry data (via B and D_{spin}). The second is the effective anisotropy K_{eff} , which they acquire from the magnetometry measurements along both the out-of-plane and in-plane directions. The third is the estimated domain wall energy, acquired from the measurement of the average domain wall width in the LTEM data, and the saturation magnetisation.

a) Given the large number of parameters (not to mention the $\gamma=1$ and $\beta=0.31$ assumptions), I find it difficult to believe the accuracy and precision of the authors' DMI value. Have the authors tried to perform some error analysis, to consider how large the error in their DMI value could be, given the error in each of these other quantities?

Reply: We agree with the referee that the experimentally calculated DMI value should be considered as an estimation. Just as the referee mentions at the end of the comment letter, it is notoriously difficult to accurately and precisely determine the DMI. For the average domain width extracted from L-TEM images, we add standard deviation in the manuscript, i.e. $0.24 \pm 0.01 \mu\text{m}$. For γ and β , we think the accurate determination of these coefficients in $\text{Fe}_{3-x}\text{GaTe}_2$ is difficult and beyond the scope of this study. But we try to choose the γ and β values as accurate as possible. To be more specific, $\beta = 0.31$ is suitable for magnets with high magnetocrystalline anisotropy. It is also found suitable for other vdW ferromagnets such as Fe_3GeTe_2 [*J. Appl. Phys.* **120**, 083903 (2016); *Nat. Commun.* **11**, 3860 (2020)] and Fe_5GeTe_2 [*Adv. Funct. Mater.* **31**, 2009758 (2021)]. Regarding the critical exponent γ , $\gamma = 1$ is widely adopted in other studies to determine the exchange stiffness constant A [*Nat. Phys.* **11**, 825-829 (2015); *Nat. Commun.* **9**, 1648 (2018)]. Based on the above values, the DMI constant is estimated to be 0.51 mJ m^{-2} . While more generally, the magnetic critical behavior of itinerant ferromagnets may diverge from the mean field model, and thus γ can be in the range from 0.7 to $4/3$ [*Nat. Phys.* **11**, 825-829 (2015); *Nat. Commun.* **9**, 1648 (2018)]. Therefore, the error of D can be further estimated according to γ . Finally, we can get $|D| = 0.56 \text{ mJ m}^{-2}$ and 0.46 mJ m^{-2} for $\gamma = 0.7$ and $4/3$, respectively.

In light of this comment, we directly indicate that the obtained DMI is an estimation in the revision. Additionally, we add more discussion about the error of $|D|$ in Supplementary Note 2:

“It is worth noting that the accurate and precise determination of DMI constant is difficult, and thus the value obtained here should be considered as an estimation. On the one hand, the effective magnetic anisotropy in the bulk and exfoliated samples may show discrepancy, i.e., K_{eff} can be thickness dependent. Considering the L-TEM sample used to extract the domain wall energy is quite thick ($\sim 179 \text{ nm}$), we suppose the magnetic properties of the bulk and this exfoliated flake are similar. Therefore, this K_{eff} discrepancy is supposed to be small. On the other hand, the values of coefficients, including β and γ , used in the calculations should be regarded as reasonable estimations. The accurate determination of these coefficients in $\text{Fe}_{3-x}\text{GaTe}_2$ is difficult and beyond the scope of this study. For instance, in general cases, the magnetic critical behavior of itinerant ferromagnets may diverge from the mean field model. The critical exponent γ is supposed to be in the range from 0.7 to $4/3$ ^{7,13}, thus the error caused by γ can be further estimated. Accordingly, we estimate that $|D|$ can be in the range of $0.46 - 0.56 \text{ mJ m}^{-2}$.”

b) Why do the authors only consider 5 data points in their Fig. S9, while they have clearly measured MvH at more temperatures (from the Fig. 1 data)?

Reply: We only consider 5 low-temperature data points because the Bloch $T^{3/2}$ law works at low temperatures. These 5 data points are collected at 4, 10, 50, 100, and 150 K. Although there are more data points measured above 150 K in Fig. 1b, they may diverge from the Bloch $T^{3/2}$ law. In new Supplementary Fig. 10, we put the original low-temperature $M(H)$ loops:

Fig. R12. Supplementary Fig. 10 | Saturation magnetization fitted by the Bloch $T^{3/2}$ law. a Out-of-plane bulk isothermal magnetization at low temperatures. **b** The Bloch $T^{3/2}$ law fitting.

c) I have further specific issues with the K_{eff} value. Firstly, the MvH loops clearly show hysteresis in Fig. 1b (is there any for the IP direction?). How is the hysteresis considered when calculating the K_{eff} value? More importantly, the value of K_{eff} has been estimated from the bulk crystal. However, it is well known that shape and surface anisotropies are also relevant. Is this K_{eff} value from the bulk crystal really representative of the K_{eff} of the exfoliated flakes? Could it be possible to acquire a more accurate value of K_{eff} directly for flakes using transport measurements (I suppose this would require performing IP field transport measurements)

Reply: The out-of-plane $M(H)$ loops indeed show hysteresis, especially at low temperatures. Whereas the hysteresis is small at 300 K, as shown in Fig. R13. For the in-plane $M(H)$ loops, the hysteresis is negligible at 300 K.

Fig. R13. Bulk out-of-plane and in-plane $M(H)$ loops measured at 300 K.

To calculate K_{eff} for hysteretic $M(H)$ loops, the hysteresis can be removed by averaging the two loop branches. Then K_{eff} can be calculated using the obtained anhysteretic curves [*J. Magn. Magn. Mater.* **116**, L1-L6 (1992); *Rep. Prog. Phys.* **59**, 1409-1458 (1996); *J. Mater. Chem. C* **11**, 4820-4829 (2023)]. Finally, the K_{eff} is estimated to be 0.30 MJ m^{-3} at 300 K. We modify the expression in Supplementary Note 2 as follows:

“For the effective magnetic anisotropy constant, it can be calculated using the area method⁹:

$$K_{\text{eff}} = \int_0^{M_s} [H_{\perp}(M) - H_{\parallel}(M)] dM, \quad (\text{S5})$$

where H_{\parallel} and H_{\perp} represent the out-of-plane and in-plane magnetic field, respectively. In practice, for hysteretic $M(H)$ loops, the hysteresis can be removed by averaging the two loop branches, and the K_{eff} can be calculated using the obtained anhysteretic curves⁹⁻¹¹.”

Besides, we agree that the effective magnetic anisotropy in bulk and exfoliated samples may show discrepancy, i.e., K_{eff} can be thickness dependent. For 2D vdW ferromagnetic materials with strong perpendicular anisotropy and near square-shaped loops, one can estimate the magnetic anisotropic energy by fitting the Stoner–Wohlfarth model [*Nat. Commun.* **9**, 1554 (2018)], which requires the angle-dependent transport measurements, just as the referee supposes. It is known that the Stoner–Wohlfarth model is a classical model which describes magnetic hysteresis of single-domain magnets. However, our L-TEM samples clearly show multi domains at 300 K. The device sample in Fig. 1c show a sheared loop at room temperature, also indicating a multi-domain state. Thus, it is questionable to use the Stoner–Wohlfarth model to extract the magnetic anisotropic energy in this situation. Maybe this model is applicable for very thin flakes at low temperatures, because at that time the samples show near square-shaped loops, just like the case in Fe_3GeTe_2 [*Nat. Commun.* **9**, 1554 (2018)]. On the other hand, the L-TEM sample that is used to estimate the domain width and domain wall energy is about 179 nm. It is quite a thick flake, thus we think the magnetic properties of the bulk and this exfoliated flake could be similar. In light of this comment, we add a note in Supplementary Note 2:

“It is worth noting that the accurate and precise determination of DMI constant is difficult, and thus the value obtained here should be considered as an estimation. On the one hand, the effective magnetic anisotropy in the bulk and exfoliated samples may show discrepancy, i.e., K_{eff} can be thickness dependent. Considering the L-TEM sample used to extract the domain wall energy is quite thick (~179 nm), we suppose the magnetic properties of the bulk and this exfoliated flake are similar. Therefore, this K_{eff} discrepancy is supposed to be small.”

7) The HAADF-STEM images in Fig. 4 are perhaps the most crucial results for the authors’ arguments. Once again, I have several/comments questions about these results

a) Due to the lower resolution of my referee files in combination with the small size of panel f, it is difficult to see, and therefore believe, this Fe site displacement result. I would like to see the authors greatly enlarge panel 4f, and add more detail. For example, could they fit the three peaks with Guassian/Lorentz functions, to more accurately determine the central position of each atom, and indicate these values much more clearly on the plot? From my own analysis of the poor resolution image using ImageJ, the displacement must be quite small, so careful analysis must be clearly shown.

Reply: We guess it is due to the image compression in the submission process. In light of this comment, we remove Fig. 4a,b (they are included in Supplementary information) to give more space to enlarge the intensity profile of the atoms. In addition, we use the Voigt function, which

is a convolution of Gaussian and Lorentzian functions, to fit the peaks. By following the referee's suggestions, the Fe_{II} displacement ($\sim 0.15 \text{ \AA}$) is illustrated more clearly in Fig. 4c (Fig. R14 below).

Fig. R14. Fig.4 Observation of off-centered Fe_{II} in $\text{Fe}_{3-x}\text{GaTe}_2$. **a** HAADF-STEM image taken along the $[100]$ zone axis. **b** Magnified image captured from **a**. **c** Intensity profile of the orange dashed lines in **b**. The profile is fitted by the Voigt function (dashed curves). The peak positions are indicated by the vertical lines. **d** HAADF-STEM images taken along the $[001]$ zone axis. Inset is a magnified image. **e** Schematic of Fe_3GaTe_2 crystal structure. The dashed rectangle indicates the unit cell. The off-centered Fe_{II} reduces the crystal symmetry.

b) I was wondering if the authors have investigated other portions of the crystal? I am curious to see if the same inversion breaking direction is seen throughout the crystal. Do the authors have any speculation as to how/why the crystal would uniformly pick one breaking of the inversion symmetry?

Reply: The cross-sectional HAADF-STEM sample in Fig. 4 is randomly cut from a bulk crystal by FIB. We try to investigate more places on that crystal. But unfortunately, as our Probe Cs-corrected STEM has some issues recently, we cannot obtain clear atomic-resolution images after many attempts. Alternatively, we perform first-principles calculations to calculate the energy difference between the parallel and anti-parallel Fe_{II} displacements at adjacent layers. It is found that, for a displacement of 0.15 \AA , the total energy of the parallelly Fe_{II} -shifted $\text{Fe}_{3-x}\text{GaTe}_2$ is 6.2

meV/u.c. lower than the anti-parallelly Fe_{II}-shifted one. This might suggest that the crystal is energetically favorable to pick one direction of Fe_{II} displacement. But it should be noted that this energy difference is quite small. In the actual situation, the defect conditions, such as Fe vacancies and/or interstitial Ga and Te atoms, in Fe_{3-x}GaTe₂ (i.e. Fe_{2.85}GaTe_{2.03} for our crystals) can be very complicated, which may change this energy difference.

c) Is it possible to use the x-ray diffraction data to perform a structural refinement, and corroborate the broken inversion symmetry of the Fe(ii) atom site? I imagine this change in symmetry should result in the appearance of small diffraction peaks, which may be forbidden in the centrosymmetric crystal structure.

Reply: Thanks for this suggestion. Actually, we didn't perform single crystal XRD experiment and structural refinement. If we are correct, the resolution of XRD is limited by the wavelength of X-ray source, e.g., the wavelength of Mo-K_α is ~0.71073 Å, which roughly leads to a maximum resolution of $\gamma/2 \approx 0.36$ Å according to the Bragg's law. Whereas the Fe_{II} displacement is about 0.15 Å. Thus, we speculate that the structural refinements based on the Fe_{II}-shifted and unshifted Fe_{3-x}GaTe₂ models will give very similar fit quality factors. Moreover, we use software to calculate the theoretical powder XRD spectra of Fe_{II}-shifted and unshifted Fe_{3-x}GaTe₂, as shown in Fig. R15. One can see that the Fe_{II}-shifted Fe_{3-x}GaTe₂ does not show additional diffraction peaks.

Fig. R15 Theoretical powder XRD spectra of Fe_{II}-shifted and unshifted Fe_{3-x}GaTe₂.

d) I am wondering if the authors utilised their DFT simulations to check the energy difference between the centrosymmetric and non-centrosymmetric crystal structures (forgetting about magnetism for the moment). Is it energetically favourable to form this structure, or is the energy cost relatively small? Any clues about why the symmetry would break in this way?

Reply: We agree with the referee that the DFT calculations may give hints to the reason of Fe_{II} displacement. For stoichiometric Fe₃GaTe₂ (i.e. without defects), our DFT calculations reveal that the total energy of the Fe_{II}-shifted Fe₃GaTe₂ ($\Delta d = 0.15 \text{ \AA}$) is 316 meV/u.c. higher than the unshifted one, meaning that it is energetically unfavorable to form the non-centrosymmetric crystal structure in stoichiometric Fe₃GaTe₂. Considering the Fe deficiency in our samples, we speculate that the observed Fe_{II} displacement might be caused by defects, such as Fe vacancies and/or interstitial Ga and Te atoms. We also try several Fe_{3-x}GaTe₂ models with very simple defects to do the simulation, but the results are not perfectly satisfactory. This may suggest that the defect conditions in Fe_{3-x}GaTe₂ can be complex. Therefore, we would like to leave this interesting topic as an open question for future studies. In light of this comment, we add discussion on page 13:

“It is worth mentioning that our first-principles calculations reveal that the stoichiometric Fe₃GaTe₂ (without defects) is energetically more favorable to form the centrosymmetric crystal structure rather than the non-centrosymmetric one. Considering the Fe deficiency in our samples, we speculate that the observed Fe_{II} displacement might be caused by defects, such as Fe vacancies and/or interstitial Ga and Te atoms, which can be complex in Fe_{3-x}GaTe₂. We would like to leave this interesting topic as an open question for future studies.”

e) The authors may wish to compare their work to another Neel-type skyrmion host, also a Polar magnet: GaV₄S₈ [I. Kesmarki et al. Nat. Mater. 14, 1116-1122 (2015)].

Reply: We thank the referee for this suggestion. We also notice other two polar magnets that can host skyrmions, i.e. VOSe₂O₅ [*Phys. Rev. Lett.* **119**, 237201 (2017)] and PtMnGa [*Adv. Mater.* **32**, 1904327 (2020)]. Accordingly, we add a comparison between these materials and Fe_{3-x}GaTe₂ on page 8:

“Even in traditional non-vdW bulk materials, high-temperature chiral skyrmions are scarce. For instance, GaV₄S₈, VOSe₂O₅, and PtMnGa are polar magnets hosting Néel-type skyrmions, whereas their skyrmion phase emerges far below room temperature^{53,55,56}.”

8) Excluding the presence of oxidised layers is a crucial point when comparing to other works in the literature. Did the authors perform any measurements to characterise the oxidisation of their samples (even the best glovebox atmosphere may lead to some oxidisation)?

Reply: We perform electron energy loss spectroscopy (EELS) measurements to examine the oxidization in hBN-covered and uncovered regions, as shown in Fig. R16 (Supplementary Fig. 12). The results reveal that the hBN layer can effectively protect Fe_{3-x}GaTe₂ from oxidization.

Fig. R16. Supplementary Fig. 12 | EELS spectra of hBN-covered and uncovered $\text{Fe}_{3-x}\text{GaTe}_2$. The oxygen K edge is observed in uncovered regions, whereas it is undetected in hBN-covered regions.

Accordingly, we add a sentence on pages 9 and 10:

“Supplementary Figure 11 shows a 130 nm-thick $\text{Fe}_{3-x}\text{GaTe}_2$ flake which is partially protected by hBN. Electron energy loss spectroscopy (EELS) measurements reveal that the hBN layer can effectively protect $\text{Fe}_{3-x}\text{GaTe}_2$ from oxidization (Supplementary Fig. 12).”

9) I’m not so familiar with the method, but does the DMI calculated from the DFT depend on the assumed spin structure (in the present case, the authors have utilised a 90 degree neighbouring spin rotation between cells). I assume that this is not the case?

Reply: Thanks for this good comment. The DFT calculated DMI constant may change with the wavelength of the cycloid in the calculation model, which has been discussed in the original paper of the real-space spin spiral method (Ref. 68 in the main text) [*Phys. Rev. Lett.* **115**, 267210 (2015)]. Whereas this change should be small when we extend the wavelength (i.e. decrease the rotation angle between neighboring spins). For instance, in the material system Pt(3)Co(1) bilayers in Ref. 68, the change of DMI constant is less than 7% when the rotation angle decreases from 90° to 45°. Nevertheless, with the reduction of rotation angle, the computational cost significantly increases because it needs a much larger supercell. Therefore, the 90° rotation angle is often used in calculations for a balance between the computational efficiency and accuracy.

Overall, I believe that the observation of Neel-type skyrmions, together with the broken inversion symmetry of the crystal structure, are already vital findings for the community, and this is already enough to warrant publication. It is not necessary to try to accurately and precisely determine the DMI value in the sample (which is notoriously difficult). Although an estimation is welcome, as long as it is clearly an estimation. For me, the most important consideration before acceptance is the careful and clear confirmation of the broken inversion symmetry in the STEM images.

Reply: We thank the referee for the positive evaluations. According to the comments, we emphasize in the revised manuscript that the calculated DMI constant is an estimation. We appreciate the referee's understanding regarding the difficulty of acquiring an accurate and precise DMI value. For the HAADF-STEM images, we enlarge the corresponding figures and fit the profile peaks with functions to extract a more accurate atom displacement. We also perform more L-TEM experiments with a series of defocus to confirm the Néel-type skyrmions.

Additionally, encouraged by the referee's comments, we perform more field- and thickness-dependent L-TEM experiments. Through a comprehensive analysis, we conclude the skyrmion stabilization mechanism. That is, the magnetic skyrmions in $\text{Fe}_{3-x}\text{GaTe}_2$ should be primarily dipolar stabilized, and the DMI is essential to guarantee the Néel-type configuration. More interestingly, another type of topological spin texture, i.e. the magnetic skyrmionium, is included and discussed. The diversity of topological spin textures and their phase transitions reveal that $\text{Fe}_{3-x}\text{GaTe}_2$ is a rich platform for room-temperature skyrmionics.

We greatly thank the referee for these constructive comments that substantially improve our manuscript.

Response to Reviewer #2

In this work by C. Zhang et. al., the authors report on the observation of Néel type skyrmions in Fe_{3-x}GaTe₂. They analyze the magnetic properties of the bulk material to determine above room temperature Curie point, as well as magnetocrystalline anisotropy along the c-axis. The LTEM data is presented which shows clearly the domains formed are of Néel type. Following a FC protocol, they are able to stabilize the skyrmion lattice which is a meta-stable phase and demonstrated by heating the sample close to Curie temperature. The origin of interfacial DMI is then presented which is shown to be due to the off-centering of the Fe(II) sites. This is confirmed through high resolution STEM imaging as well as DFT. Overall this work is very well done, and comprehensively shows the formation, and origin of Néel type skyrmions in this material at RT. The paper is recommended for publication in Nat. Comm. with some revisions as noted below.

1) Can the authors comment about the reason for the change in the slope of the magnetization curve? Is it caused by defects impeding the movement of the domains or is there a secondary phase contribution?

Reply: We thank the referee for reviewing our manuscript and giving positive evaluations. The change in the slope of the magnetization curve is caused by the irreversible process corresponding to the nucleation and annihilation of domains [Phys. Rev. B 70, 224434 (2004)]. In the saturation process, the stripe domains fragment into short chains and skyrmions, leading to a gradual increase of magnetization until saturation. Whereas in the demagnetization process, the single-domain state remains until the labyrinth domain pops up, leaving a steep slope on the magnetization curve. A similar and exaggerated example is shown below, which is reproduced from [Phys. Rev. B 105, 064426 (2022)]:

Fig. R17. a Kerr microscopy images: domain structure of a thin lamella of Nd₂Fe₁₄B (2 μm thick), observed by Kerr microscopy on of the (001) surface with a magnetic field applied parallel to the *c* axis (perpendicular to the observed plane) at room temperature. The magnetic fields corresponding to the domain patterns in A–I can be taken from the plot in b. **b** Magnetization curve in a thin lamella of Nd₂Fe₁₄B. Reproduced from [Phys. Rev. B 105, 064426 (2022)].

In light of this comment, we rephrase the description on pages 4 and 5 as follows:

“...It is interesting to note that the out-of-plane magnetization curves below T_c show two distinct slopes (i.e. the steep and slanted slopes), which is caused by the irreversible process corresponding to the nucleation and annihilation of domains⁴⁶. In the saturation process, the stripe domains fragment into short chains and skyrmions, leading to a gradual increase of magnetization until saturation. Whereas in the demagnetization process, the single-domain state remains until the labyrinth domain pops up, leaving a steep slope on the magnetization curve...”

2) The procedure to obtain skyrmion lattice is mis-represented by calling it zero-field skyrmions. This method of generating skyrmions has been pretty well established, and as the authors mention, they are nucleated only during FC process where a field is necessary. After the removal of the field at room temperature, the skyrmions still exist because they form a metastable state. This should not be referred to as zero-field skyrmions.

Reply: After careful consideration, we agree with the referee that it is not rigorous to call them zero-field skyrmions, although some papers do so [*Nano Lett.* 2018, 18, 7777–7783; *NPG Asia Materials* (2022) 14:74]. Because it is still necessary to apply an external field in the FC process. In the revised manuscript, we remove the word “zero-field skyrmions” and call them “metastable skyrmions at zero magnetic field”.

3) Can the authors comment on the origin of the Fe(II) ion shift i.e. the origin for it? Is it due to the off-stoichiometry of the compound i.e. being iron deficient, or is there another reason. And what could be ways of controlling this behavior?

Reply: We try to use DFT calculations to find hints for the Fe_{II} displacement. It is revealed that the stoichiometric Fe₃GaTe₂ (without defects) is energetically more favorable to form the centrosymmetric crystal structure rather than the non-centrosymmetric one. Considering the Fe deficiency in our samples, we speculate that the observed Fe_{II} displacement might be caused by defects, such as Fe vacancies and/or interstitial Ga and Te atoms. We also try several Fe_{3-x}GaTe₂ models with very simple defects to do the simulation, but the results are not perfectly satisfactory. This may suggest that the defects conditions in Fe_{3-x}GaTe₂ can be complex. We further speculate that the Fe_{II} displacement should be controllable if one can precisely regulate the defects in Fe_{3-x}GaTe₂. But in practice, we find it is not easy to get perfect stoichiometric Fe₃GaTe₂ crystals using CVT growth method. Therefore, we would like to leave this interesting topic as an open question for future studies. Accordingly, we add discussion on page 13:

“It is worth mentioning that our first-principles calculations reveal that the stoichiometric Fe₃GaTe₂ (without defects) is energetically more favorable to form the centrosymmetric crystal structure rather than the non-centrosymmetric one. Considering the Fe deficiency in our samples, we speculate that the observed Fe_{II} displacement might be caused by defects, such as Fe vacancies and/or interstitial Ga and Te atoms, which can be complex in Fe_{3-x}GaTe₂. We would like to leave this interesting topic as an open question for future studies.”

Minor: Definition of in-plane and out-of-plane is incorrect – $H // c$, and $H \perp c$ on last line page 3.

Reply: In our definition, c means the c axis rather than the c plane. Therefore, $H \perp c$ and $H // c$ represent the in-plane and out-of-plane field configuration, respectively. To express this more clearly, we modify the description on page 4 as follows:

“The $(00l)$ peaks in the θ - 2θ X-ray diffraction spectrum (Supplementary Fig. 2) imply that the crystallographic c axis is along the out-of-plane direction of the crystal. Figure 1a shows the temperature-dependent magnetization of $\text{Fe}_{3-x}\text{GaTe}_2$ under the in-plane ($H \perp c$, blue curves) and out-of-plane ($H // c$, red curves) magnetic field configurations, where $H \perp c$ and $H // c$ indicates the field is perpendicular and parallel to the c axis, respectively.”

Response to Reviewer #3

This work by C. Zhang et. al. reports on the realization of room-temperature chiral skyrmion lattice in a van der Waals ferromagnet $\text{Fe}_{3-x}\text{GaTe}_2$. Given that the $\text{Fe}_{3-x}\text{GaTe}_2$ is known to be a centrosymmetric structure, the observed chiral skyrmion lattice can be a surprising. The authors attribute the chiral skyrmion to unexpected displacement of Fe atoms, leading to the Dzyaloshinskii-Moriya interaction (DMI). While their results are interesting, and the claims are well supported by robust experimental and theoretical evidences, I am a bit negative to the publication of this work in Nature Communications for the following reasons below.

1) First, room-temperature chiral Neel skyrmions in 2D van der Waals material is already reported in 2022 [ref. 35]. While the present work emphasizes that their skyrmion is stabilized above-room temperature, I think this is not surprising progress because anyhow room-temperature skyrmion means that the skyrmion can be observed above room-temperature. This is just matter of how wide the temperature range is.

Reply: We thank the referee for reviewing our manuscript. We use “above-room-temperature” because we want to emphasize that the skyrmion lattice phase temperature in Fe_3GaTe_2 is beyond room temperature and higher than other known van der Waals materials. It is important because high-temperature skyrmion materials have a superior application potential. In the paper mentioned by the referee [*Sci. Adv.* **8**, eabm7103 (2022)], the Néel skyrmion lattice at zero magnetic field in Co-doped Fe_5GeTe_2 can survive up to 312 K. In our work, we demonstrate that $\text{Fe}_{3-x}\text{GaTe}_2$ possesses a record-high critical temperature (between 330 to 340 K) of the Néel skyrmion lattice state at zero magnetic field among all known van der Waals magnets. It is worth mentioning that, in current-driven skyrmion motion, Joule heating is inevitable because a large current density is required to overcome the pinning and achieve high-speed motion, which can cause a temperature rise of several tens of kelvin [*Nat. Electron.* **3**, 30-36 (2020)]. Therefore, skyrmion materials with a skyrmion phase temperature just reaching room temperature may not be able to function as practical devices at room temperature. The observation of above-room-temperature high-density chiral skyrmion lattice at zero magnetic field in $\text{Fe}_{3-x}\text{GaTe}_2$ opens promising perspectives for developing novel spintronic devices based on 2D magnets.

Similarly, in a pioneer work of Fe_3GaTe_2 [*Nat. Commun.* **13**, 5067 (2022)], the authors used “above-room-temperature” in the title to emphasize the high-temperature ferromagnetism. In Table R1, we list several recent papers that use “above-room-temperature” in the titles, including one that studies Co-doped Fe_5GeTe_2 .

Table R1

Title	Materials	T_C	Reference
Above-room-temperature strong intrinsic ferromagnetism in 2D van der Waals Fe_3GaTe_2 with large perpendicular magnetic anisotropy	Fe_3GaTe_2	367 K	Nat. Commun. 13 , 5067 (2022)
Electronic Structure of Above-Room-Temperature van der Waals Ferromagnet Fe_3GaTe_2	Fe_3GaTe_2	380 K	Nano Lett. (2023) DOI: 10.1021/acs.nanolett.3c03203
Above-Room-Temperature Ferromagnetism in Thin van der Waals Flakes of Cobalt-Substituted Fe_5GeTe_2	Co-doped Fe_5GeTe_2	328 K	ACS Appl. Mater. Interfaces 15 , 3287-3296 (2023)
Above-Room-Temperature Ferromagnetism in Copper-Doped Two-Dimensional Chromium-Based Nanosheets	Cu-doped Cr_7Te_8	315 K	ACS Nano (2023) DOI: 10.1021/acsnano.3c08998
Ferromagnetism above Room Temperature in Two Intrinsic van der Waals Magnets with Large Coercivity	MnSiTe_3 MnGeTe_3	378 K 349 K	Nano Lett. 23 , 11226–11232 (2023)

Based on the above arguments, we think the observation of above-room-temperature skyrmion lattice at zero magnetic field is a valuable progress compared to previous studies. We also think the use of “above-room-temperature” in the manuscript is appropriate. To highlight this novelty more clearly, we add discussion on page 8:

“It is worth mentioning that, in current-driven skyrmion motion, Joule heating is inevitable because a large current density is required to overcome the pinning and achieve high-speed motion, which can cause a temperature rise of several tens of kelvin⁶². Therefore, skyrmion materials with a skyrmion phase temperature just reaching room temperature may not be able to function as practical devices at room temperature. The observation of above-room-temperature high-density chiral skyrmion lattice at zero magnetic field in $\text{Fe}_{3-x}\text{GaTe}_2$ opens promising perspectives for developing novel spintronic devices based on 2D magnets.”

2) Second, the controllability of the DMI reported in the manuscript is quite skeptical. In the introduction, the authors listed up several previous results which report ways to induce inversion symmetry breaking and thus the DMI, and the authors pointed out that the previous reports still have limitations because these approaches are challenging to control. However, I am not sure the DMI in proposed this work is superior in terms of controllability, meaning that this work still has the same limitation.

Reply: In the introduction, we mention the inversion symmetry breaking and chiral skyrmions can possibly be induced by self-intercalation, doping, and constructing Janus 2D magnets, whereas these methods have limitations. Maybe the word “control” we use is misleading. We have no intention to say we can control the DMI in this work. In fact, we want to express that the above

approaches are not easy to achieve. Firstly, constructing Janus 2D magnets only remains at the theoretical stage. Secondly, though Néel skyrmions are observed in self-intercalated chromium telluride [*Nat. Commun.* **13**, 3965 (2022)], more studies report Bloch-type achiral skyrmions and biskyrmions in this material system even with similar intercalate concentrations [*Adv. Mater.* **35**, 2205967 (2023); *ACS Nano* **16**, 13911-13918 (2022); *Mater. Today* **57**, 66-74 (2022)], indicating that it is not easy to realize asymmetric intercalation. Thirdly, $(\text{Fe}_{0.5}\text{Co}_{0.5})_5\text{GeTe}_2$ is a remarkable material that can host Néel skyrmion lattice up to 312 K at zero magnetic field [H. Zhang, *et al.*, *Sci. Adv.* **8**, eabm7103 (2022); P. Meisenheimer, *et al.*, *Nat. Commun.* **14**, 3744 (2023)]. To obtain the high-temperature ferromagnetic order, the key factors are precisely regulating the Co-doping concentration at $\sim 50\%$ and acquiring the AA' -type layer stacking [*Phys. Rev. Mater.* **6**, 044403 (2022); *Phys. Rev. Mater.* **4**, 074008 (2020)]. Nevertheless, obtaining a high doping concentration is not easy, the authors of [*Appl. Phys. Lett.* **116**, 202402 (2020)] claim that crystal growth for Co concentration higher than 44% are unsuccessful. Furthermore, the stacking mechanism is so far unclear, and recent studies also observe the antiferromagnetic order in $(\text{Fe}_{0.5}\text{Co}_{0.5})_5\text{GeTe}_2$ due to the existence of AA -type stacking mode [*arXiv:2308.13408*, (2023)]. Thus, $(\text{Fe}_{0.5}\text{Co}_{0.5})_5\text{GeTe}_2$ crystals with pure ferromagnetic order are not easy to synthesize. Based on the above facts, we mention these approaches are not easy to achieve. On the contrary, $\text{Fe}_{3-x}\text{GaTe}_2$ is easy to synthesize, and there have been papers reporting the robust ferromagnetism in $\text{Fe}_{3-x}\text{GaTe}_2$, although it was discovered only one year ago. In light of this comment, we rephrase the corresponding sentences on page 3:

“Although the inversion symmetry breaking and chiral skyrmions can possibly be induced by, for example, self-intercalation^{34,35}, doping¹⁷, and constructing Janus 2D magnets³⁶, these approaches are not easy to achieve. For instance, though Néel skyrmions are observed in self-intercalated chromium telluride³⁴, more studies report Bloch-type achiral skyrmions and biskyrmions even with similar intercalate concentrations^{28,29,37}, implying that the asymmetric intercalation is challenging. $(\text{Fe}_{0.5}\text{Co}_{0.5})_5\text{GeTe}_2$ is a remarkable material that can host Néel skyrmion lattice up to 312 K at zero magnetic field^{17,38}. To obtain the high-temperature ferromagnetic order, the key factors are precisely regulating the Co-doping concentration at $\sim 50\%$ and acquiring the AA' -type layer stacking^{39,40}. Nevertheless, the stacking mechanism is so far unclear, and recent studies also observe the antiferromagnetic order in $(\text{Fe}_{0.5}\text{Co}_{0.5})_5\text{GeTe}_2$ due to the existence of AA -type stacking mode⁴¹”

3) Third, whereas the origin of the DMI in the $\text{Fe}_{3-x}\text{GaTe}_2$ is firmly confirmed, the manuscript lacks information regarding why the FeII atoms are displaced and how the direction of the displacement is determined in the crystal are missing in the manuscript. I think these are also important factors if one to develop skyrmion-devices using 2D materials with high controllability.

Reply: Thanks for this comment. On page 10 in the main text, we wrote “Compared to the formerly reported $P6_3/mmc$ structure, an unexpected vertical displacement ($\sim 0.15 \text{ \AA}$) is observed at the FeII sites (orange balls) in both upper and lower layers of the unit cell, as shown in Fig. 4b,c. Whereas

in the ab plane (Fig. 4d), no atomic displacement is detected, indicating that the Fe_{II} atoms only shift along the c axis.”. This is how we determine the displacement direction.

Regarding the reason for the Fe_{II} displacement, we try to find hints from DFT calculations. It is revealed that the stoichiometric Fe_3GaTe_2 (without defects) is energetically more favorable to form the centrosymmetric crystal structure rather than the non-centrosymmetric one. Considering the Fe deficiency in our samples, we speculate that the observed Fe_{II} displacement might be caused by defects, such as Fe vacancies and/or interstitial Ga and Te atoms. We also try several $\text{Fe}_{3-x}\text{GaTe}_2$ models with very simple defects to do the simulation, but the results are not perfectly satisfactory. This suggests that the defect conditions in $\text{Fe}_{3-x}\text{GaTe}_2$ might be complex. Therefore, we would like to leave this interesting topic as an open question for future studies. We think the main aim of the present work is to present the observation of high-temperature Néel skyrmions in $\text{Fe}_{3-x}\text{GaTe}_2$ and explore the DMI mechanism based on experimental evidence. In light of this comment, we add discussion on page 13:

“It is worth mentioning that our first-principles calculations reveal that the stoichiometric Fe_3GaTe_2 (without defects) is energetically more favorable to form the centrosymmetric crystal structure rather than the non-centrosymmetric one. Considering the Fe deficiency in our samples, we speculate that the observed Fe_{II} displacement might be caused by defects, such as Fe vacancies and/or interstitial Ga and Te atoms, which can be complex in $\text{Fe}_{3-x}\text{GaTe}_2$. We would like to leave this interesting topic as an open question for future studies.”

4) For the reasons above, I am hesitating to recommend the publication of this manuscript in Nature Communications.

Reply: First of all, we thank the referee for insightful questions. The referee raises concerns about the novelty of this work, mainly because room-temperature Néel skyrmions have been reported in Co-doped Fe_5GeTe_2 . We believe that the observation of above-room-temperature (up to 330–340 K) high-density skyrmion lattice at zero magnetic field is a valuable progress compared to previous studies (e.g. up to 312 K in Co-doped Fe_5GeTe_2). In current-driven skyrmion motion, Joule heating is inevitable because a large current density is required to overcome the pinning and achieve high-speed motion, which can cause a temperature rise of several tens of kelvin. Therefore, skyrmion materials with a skyrmion phase temperature just reaching room temperature may not be able to function as practical devices at room temperature. In this perspective, our findings demonstrate that $\text{Fe}_{3-x}\text{GaTe}_2$ has superior application potential than other known van der Waals magnets. The above arguments have been added in the revised manuscript on page 8. On the other hand, our studies for the first time reveal the underlying physical mechanism of DMI in $\text{Fe}_{3-x}\text{GaTe}_2$, which is important but lacking in previous studies. Based on these facts, Referee #1 gave positive evaluations to our work, “Overall, I believe that the observation of Néel-type skyrmions, together with the broken inversion symmetry of the crystal structure, are already vital findings for the community, and this is already enough to warrant publication”. Referee #2 also mentioned

“Overall this work is very well done, and comprehensively shows the formation, and origin of Néel type skyrmions in this material at RT”.

In addition, encouraged by Referee#1’s comments, we have performed field- and thickness-dependent L-TEM experiments (Supplementary Figs. 19 and 20). Through a comprehensive analysis, we find that the magnetic skyrmions in $\text{Fe}_{3-x}\text{GaTe}_2$ are primarily dipolar stabilized, and the DMI is essential to guarantee the Néel-type configuration (see the discussion on pages 13 and 14 in the main text). This skyrmion stabilization mechanism is distinct from our knowledge in traditional chiral helimagnets. More interestingly, another type of topological spin texture, i.e. the magnetic skyrmionium, is observed (see Supplementary Fig. 6 and the discussion on pages 5 and 6 in the main text). The skyrmionium can even transform into a skyrmion when the magnetic field increases. The diversity of topological spin textures and their phase transitions reveal that $\text{Fe}_{3-x}\text{GaTe}_2$ is a rich platform for room-temperature skyrmionics. Therefore, we believe our manuscript is substantially improved after the revision.

Reviewers' Comments:

Reviewer #1:

Remarks to the Author:

The authors have done an excellent job responding to my queries and comments, including performing additional experiments. The majority of my questions have now been answered, and I really commend the authors' on their outstanding efforts to improve the manuscript.

However, at least for me, there remains outstanding questions about the reported noncentrosymmetry of the observed structure. I think this is really the key part of the results in this manuscript, and for me to recommend publication, I feel all efforts should be made to demonstrate that the overall crystal structure is indeed that of a polar compound, rather than the expected centrosymmetric. If the authors can do this, it would be a significant advance for the topic of 2D magnets, since in the literature, most of the FGT compounds share a similar controversy about the possible origin of the interfacial-like DMI. I share more detailed feedback below.

As pointed out by another reviewer, it seems extraordinary that the entire crystal could form with this Fe(2) site displaced in the same direction. The authors' own DFT calculations show that this should be energetically unfavourable. We know from the present LTEM data, that if the authors are claiming all the skyrmions have the same Neel-type chirality, and there really is a DMI which is the origin of this behaviour, then this Fe(2) displacement ordering should be the same across many square micrometers (not only the small region the authors' have seen in their electron microscopy data). This is what I want to see the authors try to demonstrate.

In their rebuttal letter, the authors claimed that the XRD pattern from both the centrosymmetric and polar crystal structures is identical. Unfortunately I do not have access to the authors' .cif files myself. However, I have taken the commonly shared Fe₃GeTe₂ .cif file, edited it to try to match the Fe₃GaTe₂ compound here, and made another version with the authors' reported noncentrosymmetric displacement (I hope it is correct, I have tried to share the files alongside this review). When I calculate the XRD powder patterns using VESTA, I do see quite significant differences. Admittedly, I could only spot one small additional peak appearing, but the relative intensity of the vast majority of the peaks does change. Below I share the superimposed patterns (see attached .pdf for image), where you can see a change in intensity for the two structures (some peaks showing the red vs grey colour behind, where the peak height has changed). A better plot could be made using the tabulated data.

Actually, if I look very closely at the authors' reported calculated XRD patterns, I can also see the very small additional peak at ~65 degrees (circled blue above) in their own plots. I recommend a log plot to better see these small changes).

I believe that these differences should be measurable even with a lab-based x-ray powder diffractometer. Using a Rietveld refinement, I believe the authors' should be able to show whether their FGT crystal better fits to the centrosymmetric or the polar structure. In particular, if the authors can observe the additional tiny peaks in their experimental XRD pattern, then this would be the ultimate proof that the sample is indeed not centrosymmetric. From experience, these matters of detailed crystal structure often come down to small details in the XRD pattern, but can make all the difference.

Reviewer #2:

Remarks to the Author:

The authors have made significant improvements to the manuscript and the data presented in this revised version. They have addressed all the concerns with sufficient supporting data. I recommend the publication of the paper in Nat. Comm.

Reviewer #3:

Remarks to the Author:

The authors have made considerable revisions in their manuscript including additional experiments. I thank the authors for their thoughtful responses. The main concern in the previous review report was whether the increase of the critical temperature from 312K to ~340K is significant and innovative progress. I agree with the author's response that obtaining a material which shows skyrmions at higher temperature is required condition for skyrmion applications as a current injection induces Joule heating. This one of the important conditions. However, another concern is that, for example, then if one achieves 2D vdW skyrmions at 360 K later, is this also significant achievement which guarantees the publication in Nature Communications? For this reason, I am not sure whether the higher temperature skyrmions are of significance. However, experimental revealing of the origin of DMI in this material is important. While, I, as one of possible readers, cannot surely recommend the publication of this work in Nature Communications, but I think it is also worth giving a chance for other readers to evaluate this work.

Response to Reviewer #1

The authors have done an excellent job responding to my queries and comments, including performing additional experiments. The majority of my questions have now been answered, and I really commend the authors' on their outstanding efforts to improve the manuscript.

However, at least for me, there remains outstanding questions about the reported noncentrosymmetry of the observed structure. I think this is really the key part of the results in this manuscript, and for me to recommend publication, I feel all efforts should be made to demonstrate that the overall crystal structure is indeed that of a polar compound, rather than the expected centrosymmetric. If the authors can do this, it would be a significant advance for the topic of 2D magnets, since in the literature, most of the FGT compounds share a similar controversy about the possible origin of the interfacial-like DMI. I share more detailed feedback below.

As pointed out by another reviewer, it seems extraordinary that the entire crystal could form with this Fe(2) site displaced in the same direction. The authors' own DFT calculations show that this should be energetically unfavourable. We know from the present LTEM data, that if the authors are claiming all the skyrmions have the same Neel-type chirality, and there really is a DMI which is the origin of this behaviour, then this Fe(2) displacement ordering should be the same across many square micrometers (not only the small region the authors' have seen in their electron microscopy data). This is what I want to see the authors try to demonstrate.

Reply: We thank the referee again for reviewing our revised manuscript. We agree with the referee that it is important to demonstrate the overall crystal has a polar structure, which would be a significant advance for the topic of 2D magnets. The referee concerns whether the Fe_{II} site shifts along the same direction in the entire crystal. To answer this question experimentally, we choose single-crystal XRD (SCXRD) to examine the overall crystal structure on millimeter scale. We also use selected-area electron diffraction (SAED) to double check the crystal structure. Combining these two techniques, we demonstrate that the Fe_n displacement takes place in the same direction across the crystal. The details will be shown in the answer to the next comment. All the modifications based on the referees' comments are highlighted in red in the revised manuscript.

In their rebuttal letter, the authors claimed that the XRD pattern from both the centrosymmetric and polar crystal structures is identical. Unfortunately I do not have access to the authors' .cif files myself. However, I have taken the commonly shared Fe₃GeTe₂ .cif file, edited it to try to match the Fe₃GaTe₂ compound here, and made another version with the authors' reported noncentrosymmetric displacement (I hope it is correct, I have tried to share the files alongside this review). When I calculate the XRD powder patterns using VESTA, I do see quite significant differences. Admittedly, I could only spot one small additional peak appearing, but the relative intensity of the vast majority of the peaks does change. Below I share the superimposed patterns (see attached .pdf for image), where you can see a change in intensity for the two structures (some

peaks showing the red vs grey colour behind, where the peak height has changed). A better plot could be made using the tabulated data.

Actually, if I look very closely at the authors' reported calculated XRD patterns, I can also see the very small additional peak at ~65 degrees (circled blue above) in their own plots. I recommend a log plot to better see these small changes).

I believe that these differences should be measurable even with a lab-based x-ray powder diffractometer. Using a Rietveld refinement, I believe the authors' should be able to show whether their FGT crystal better fits to the centrosymmetric or the polar structure. In particular, if the authors can observe the additional tiny peaks in their experimental XRD pattern, then this would be the ultimate proof that the sample is indeed not centrosymmetric. From experience, these matters of detailed crystal structure often come down to small details in the XRD pattern, but can make all the difference.

Reply: We deeply appreciate the referee's careful review and valuable comments. After careful deliberation, we decide to use single-crystal XRD (SCXRD) to examine the crystal structure. As we don't have many crystals in stock right now, we are afraid that we cannot obtain enough volume of powders to perform the powder XRD (PXRD) even if we grind all the crystals. In principle, SCXRD can give more information than the PXRD. In some cases, different space groups may give similar goodness of fit (GOF) during the refinement, and then selected-area electron diffraction (SAED) is helpful to identify the correct structure.

As we know, Fe_3GaTe_2 was considered centrosymmetric with $P6_3/mmc$ space group [*Nat. Commun.* **13**, 5067 (2022)]. The vertical displacements of Fe_{II} along the same direction in both upper and lower layers of the unit cell can break the centrosymmetry and lead to $P6_3mc$ space group. Note that here the displacements are assumed to be identical in the upper and lower layers. However, according to the HAADF-STEM results (see Fig. 4c and new Supplementary Fig. 17), the displacement values can be unequal in the two layers, which might be caused by the nonuniformity of local defect conditions. Actually, we noticed this unequal displacement in our original manuscript, but unfortunately we ignored it when we defined the space group. The small unequal displacements further lower the symmetry to the subgroup $P3m1$. Therefore, in the SCXRD experiment, we use $P6_3/mmc$, $P6_3mc$, and $P3m1$ these three space groups to perform the structure refinement, which lead to GOF of 1.376, 1.305, and 1.116, respectively. Thus, $P3m1$ gives the best fit quality. The refined crystallography data are summarized in Supplementary Tables 1 & 2. We use this refined structure to simulate the corresponding HAADF-STEM images and the results are consistent with the experimental observations, as shown in Supplementary Fig. 18.

Moreover, we perform selected-area electron diffraction (SAED) on $\text{Fe}_{3-x}\text{GaTe}_2$ lamellar samples to double check the crystal structure. It is known that for $P6_3/mmc$ and $P6_3mc$ space groups, the reflection condition should satisfy (hhl) , where l is an even number (i.e. $l = 2n$ and n is an integer) [Hahn, T. International Tables for Crystallography, Volume A: Space Group Symmetry

(2006)]. Whereas $P3m1$ has no such extinction rule. Supplementary Fig. 19 displays the SAED patterns taken along [100] and [210] zone axes, in which the (001), (003), and (005) reflection spots are unambiguously identified, indicating that $\text{Fe}_{3-x}\text{GaTe}_2$ should belong to $P3m1$ rather than $P6_3/mmc$ or $P6_3mc$. Thus, the HAADF-STEM, SCXRD, and SAED results are consistent with each other. The schematic of $\text{Fe}_{3-x}\text{GaTe}_2$ crystal structure is illustrated in Fig. 4e. The off-centered Fe_{II} atoms lower the crystal symmetry, making $\text{Fe}_{3-x}\text{GaTe}_2$ a polar metal that belongs to the non-centrosymmetric space group $P3m1$ (point group C_{3v}).

In the revised manuscript, the SCXRD and SAED results are shown in Supplementary Tables 1 & 2, and Supplementary Fig. 19, respectively. Supplementary Fig. 17 is added to show the unequal displacement. The simulated HAADF-STEM images based on the refined structure are displayed in Supplementary Fig. 18. We also add more discussions on pages 10 and 11 in the main text:

“Compared to the formerly reported $P6_3/mmc$ structure⁶⁵, unexpected vertical displacements are observed at Fe_{II} sites (orange balls) in both upper and lower layers of the unit cell and they are along the same direction, as shown in Fig. 4b, c and Supplementary Fig. 17. Whereas in the ab plane (Fig. 4d), no atomic displacement is detected, indicating that the Fe_{II} atoms only shift along the c axis. The vertical displacements of Fe_{II} , assuming their values are identical in the two layers, can break the centrosymmetry and lead to $P6_3mc$ space group. Nevertheless, a small discrepancy of the displacements between the two layers can further lower the symmetry to the subgroup $P3m1$. Our HAADF-STEM results support such discrepancy (see Fig. 4c and Supplementary Fig. 17).

To further determine the crystal structure on a larger scale, we perform single-crystal X-ray diffraction (SCXRD) using a millimeter-sized crystal. Based on the aforementioned discussion, we use these three space groups ($P6_3/mmc$, $P6_3mc$, and $P3m1$) to perform the structure refinement, which lead to goodness of fit of 1.376, 1.305, and 1.116, respectively. Thus, $P3m1$ provides the best fit quality. The refined crystallography data are summarized in Supplementary Tables 1 & 2. The simulated HAADF-STEM images based on the refined structure are consistent with the experimental observations, as shown in Supplementary Fig. 18. Moreover, we perform selected-area electron diffraction (SAED) on $\text{Fe}_{3-x}\text{GaTe}_2$ lamellar samples along [100] and [210] zone axes. It is known that for $P6_3/mmc$ and $P6_3mc$ space groups, the reflection condition should satisfy (hhl) , where l is an even number (i.e. $l = 2n$ and n is an integer)⁶⁶, however $P3m1$ has no such extinction rule. In Supplementary Fig. 19, the (001), (003), and (005) reflection spots are unambiguously identified, indicating that $\text{Fe}_{3-x}\text{GaTe}_2$ should belong to $P3m1$ rather than $P6_3/mmc$ or $P6_3mc$. Thus, the SAED patterns are consistent with the HAADF-STEM and SCXRD results. The schematic of $\text{Fe}_{3-x}\text{GaTe}_2$ crystal structure is illustrated in Fig. 4e. The off-centered Fe_{II} atoms lower the crystal symmetry, making $\text{Fe}_{3-x}\text{GaTe}_2$ a polar metal that belongs to the non-centrosymmetric space group $P3m1$ (point group C_{3v}).”

Supplementary Table 1 | Summary of the SCXRD data and structure refinement.

Chemical formula	Fe _{2.78} GaTe ₂
Formula weight (g mol ⁻¹)	480.34
Temperature (K)	273(2)
Crystal system	trigonal
Space group	P3m1 (No. 156)
a (Å)	4.0826(3)
b (Å)	4.0826(3)
c (Å)	16.1465(16)
α (°)	90
β (°)	90
γ (°)	120
Volume (Å ³)	233.07(4)
Z	2
Density ρ_{calc} (g cm ⁻³)	6.845
Absorption coefficient μ (mm ⁻¹)	26.286
F(000)	415
Radiation	Mo K α ($\lambda = 0.71073$ Å)
θ range for data collection (°)	2.52 to 28.27
Index ranges	$-5 \leq h \leq 5, -5 \leq k \leq 5, -21 \leq l \leq 21$
Reflections collected	2369
Independent reflections	558 [$R_{\text{int}} = 0.0467, R_{\text{sigma}} = 0.0433$]
Completeness	98.9%
Structure solution technique	direct methods
Structure solution program	SHELXT
Refinement method	Full-matrix least-squares on F^2
Refinement program	SHELXT
Function minimized	$\Sigma w(F_o^2 - F_c^2)^2$
Data/restraints/parameters	558/1/38
Goodness-of-fit on F^2	1.116
Final R indexes [$I > 2\sigma(I)$]	$R_1 = 0.0600, wR_2 = 0.1667$
Final R indexes [all data]	$R_1 = 0.0624, wR_2 = 0.1720$
Weighting scheme	$w=1/[\sigma^2(F_o^2) + (0.1259P)^2 + 0.8545P]$ where $P = (F_o^2 + 2F_c^2)/3$
Absolute structure parameter	0.4(2)
Largest diff. peak/hole ($e\text{Å}^{-3}$)	5.970/-3.541
R.M.S. deviation from mean ($e\text{Å}^{-3}$)	0.607

Supplementary Table 2 | Atomic coordinates of $\text{Fe}_{3-x}\text{GaTe}_2$. The equivalent isotropic atomic displacement parameter U_{eq} is defined as 1/3 of the trace of the orthogonalised U_{ij} tensor.

Atom	x	y	z	U_{eq} (\AA^2)	Occupancy
Fe1	0.333333	0.666667	0.2546(6)	0.015(2)	0.8509
Fe2	0.0	0.0	0.3270(6)	0.0132(19)	0.8912
Fe3	0.0	0.0	0.8280(6)	0.0164(19)	1
Fe4	0.0	0.0	0.1741(8)	0.026(3)	1
Fe5	0.0	0.0	0.6748(8)	0.018(3)	0.9032
Fe6	0.666667	0.333333	0.7543(6)	0.0126(16)	0.92
Ga1	0.333333	0.666667	0.7476(7)	0.0294(19)	1
Ga2	0.666667	0.333333	0.2468(7)	0.0304(18)	1
Te1	0.333333	0.666667	0.40883(19)	0.0190(8)	1
Te2	0.333333	0.666667	0.0909(2)	0.0240(10)	1
Te3	0.666667	0.333333	0.90860(19)	0.0168(8)	1
Te4	0.666667	0.333333	0.5907(2)	0.0229(10)	1

Fig. R2. Supplementary Fig. 17 | Displacement of Fe_{II} in the upper layer of the unit cell. a The same HAADF-STEM image in Fig. 4b in the main text. **b** Intensity profile of the orange dashed lines in **a**. The profile is fitted by the Voigt function (dashed curves). The peak positions are indicated by the vertical lines.

Fig. R3. Supplementary Fig. 18 | Comparison of the simulated and experimental HAADF-STEM images. The left, middle, and right column shows the simulated $P6_3/mmc$, simulated $P3m1$, and experimental images, respectively. The black dashed lines denote the unit cell. The simulated HAADF-STEM images are acquired by using the Dr. Probe software²².

Fig. R4. Supplementary Fig. 19 | SAED patterns of $Fe_{3-x}GaTe_2$ lamellar samples. The patterns are taken along the [100] (a) and [210] (b) zone axes.

It should be noted that the lowering of symmetry from $P6_3mc$ to $P3m1$ does not change the DMI mechanism in $\text{Fe}_{3-x}\text{GaTe}_2$. The difference is that the Fe_{II} displacements in the upper and lower layers are not equal in $P3m1$, whereas they are still along the same direction. Thus, the net DMI constant contributed by each layer will not cancel each other out. Additionally, the C_{6v} and C_{3v} point groups share the same second-order DMI tensor [*EPL* **140**, 46003 (2022)], which is shown as Eq. (1) in the main text. This DMI tensor essentially ensures the existence of interfacial-type DMI. Thus, the conclusion of the manuscript does not change.

In the revised manuscript, we keep the results of Fig. 5b, though they are calculated based on the assumption that the Fe_{II} displacements in the upper and lower layers are identical. Because there are too many possibilities if we consider the unequal displacements. We add a note in the caption of Fig. 5: “Note that the Fe_{II} displacements in the upper and lower layers are set to the same value for simplicity”. Additionally, we show the results for the SCXRD-refined structure in Fig. 5c. The layer-resolved SOC energy difference ΔE_{SOC} mainly emerges at the Te sublayers, which is a characteristic feature of the Fert–Lévy model. The overall Fe_{II} displacements determined by SCXRD are 0.075 and 0.076 Å in the two layers, leading to a DMI constant of $|D|=0.50 \text{ mJ m}^{-2}$, which is comparable to the experimentally estimated one (i.e. 0.51 mJ m^{-2} at room temperature). The new Fig. 5 is shown below. The corresponding discussion is revised as follows on page 13:

“The DFT-calculated DMI constant as a function of Fe_{II} displacement is displayed in Fig. 5b. **Note that here the Fe_{II} displacements in the upper and lower layers are set to the same value for simplicity.** The negative D indicates a preferred CW chirality, and its magnitude increases monotonically with the displacement value. Considering that the DMI is essentially derived from the spin-orbit coupling (SOC) effect, the layer-resolved SOC energy difference, ΔE_{SOC} , is calculated to help anatomize the DMI mechanism. When Fe_{II} is not shifted, the ΔE_{SOC} in each $\text{Fe}_{3-x}\text{GaTe}_2$ monolayer is symmetrically distributed but with opposite signs (Supplementary Fig. 20a), leading to a negligible DMI, which is consistent with our prior analysis. **With the emergence of Fe_{II} displacement, the ΔE_{SOC} distribution becomes asymmetric and a net ΔE_{SOC} can be obtained.** Note that the calculated d is negative, thus a negative ΔE_{SOC} corresponds to a positive contribution to the DMI. Large SOC energy difference associated to DMI emerges at the Te sublayers, which is a characteristic feature of the Fert–Lévy model^{42,43}, suggesting that the DMI mainly stems from the heavy elements through the $\text{Fe}_{\text{II}}\text{--Te--Fe}_{\text{II}}$ triplet. **Figure 5c shows the results for the SCXRD-refined structure. The overall Fe_{II} displacements determined by SCXRD are 0.075 and 0.076 Å in the two layers, leading to a DMI constant of $|D|=0.50 \text{ mJ m}^{-2}$, which is comparable to the experimentally estimated one (i.e. 0.51 mJ m^{-2} at room temperature). Therefore, the first-principles calculations demonstrate that the observed off-centered Fe_{II} can indeed induce an effective DMI via the Fert–Lévy mechanism, making $\text{Fe}_{3-x}\text{GaTe}_2$ an intrinsic 2D topological ferromagnet.”**

Fig. R5. Fig. 5 | Anatomy of DMI in $\text{Fe}_{3-x}\text{GaTe}_2$. **a** Clockwise and counter-clockwise spin spiral models used in the DFT calculation. For clarity, only Te and Fe_{II} atoms are shown. **b** Calculated microscopic and micromagnetic DMI parameters (d and D) as a function of Fe_{II} displacement. Note that the Fe_{II} displacements in the upper and lower layers are set to the same value for simplicity. **c** Schematic of spin structure and layer-resolved SOC energy difference ΔE_{SOC} calculated using the SCXRD-refined crystal structure.

Reviewers' Comments:

Reviewer #1:

Remarks to the Author:

The authors have demonstrated the overall change in space group and the polar structure required for the suggested Neel-type DMI in this compound. The new X-ray and electron diffraction data seems to conclusively confirm this point. This is a very important result, possibly also resolving outstanding questions on the related Fe_3GeTe_2 compound. I would now recommend the article for publication.